# THE UNCANNY VALLEY: EXPLORING ADVERSARIAL ROBUSTNESS FROM A FLATNESS PERSPECTIVE

## ABSTRACT

Flatness of the loss surface not only correlates positively with generalization, but is also related to adversarial robustness, since perturbations of inputs relate non-linearly to perturbations of weights. In this paper, we empirically analyze the relation between adversarial examples and relative flatness with respect to the parameters of one layer. We observe a peculiar property of adversarial examples in the context of relative flatness: during an iterative first-order white-box attack, the flatness of the loss surface measured around the adversarial example *first* becomes sharper until the label is flipped, but if we keep the attack running, it runs into a flat *uncanny valley* where the label remains flipped. In extensive experiments, we observe this phenomenon across various model architectures and datasets, even for adversarially trained models. Our results also extend to large language models (LLMs), but due to the discrete nature of the input space and comparatively weak attacks, adversarial examples rarely reach truly flat regions. Most importantly, this phenomenon shows that flatness alone cannot explain adversarial robustness unless we can also guarantee the behavior of the function around the examples. We therefore theoretically connect relative flatness to adversarial robustness by bounding the third derivative of the loss surface, underlining the need for flatness in combination with a low global Lipschitz constant for a robust model.

## 1 INTRODUCTION

Despite the remarkable performance of modern deep learning models, their vulnerability to adversarial attacks, i.e., small crafted perturbations in the input that fool a model into changing its prediction to an incorrect label (Szegedy et al., 2014; Carlini & Wagner, 2017a), undermine the trust in the reliability of these models. Although there has been significant progress in developing methods to enhance adversarial robustness, the underlying mechanisms that dictate the susceptibility of a model to such perturbations are still not fully understood. One promising approach is the study of flatness of the loss surface. A lot of previous work demonstrated a positive correlation between flatness of the loss surface and the generalization ability of a model (Hochreiter & Schmidhuber, 1994; Keskar et al., 2016; Jiang et al., 2019). Flat minima in the loss landscape are thought to be indicative of better generalizing models, as they suggest that small changes in the parameters space do not significantly affect the loss of a model. This concept has led to the hypothesis that flatness is also related to adversarial robustness. While flatness measured with respect to inputs obviously relates to adversarial robustness (Moosavi-Dezfooli et al., 2019), it is not clear how flatness with respect to parameters connects to it (Wu et al., 2020; Kanai et al., 2023).

In this paper, we explore the relationship between flatness of the loss surface with respect to parameters and adversarial examples. We empirically investigate this connection by analyzing the behavior of the loss surface during iterative first-order white-box attacks. Our findings reveal an intriguing phenomenon, of which we give an example in Fig. 1. The loss surface initially becomes sharper as the attack progresses and the prediction is flipped. However, if the attack is continued, the adversarial example often moves into a flat region of the loss surface, which we term *uncanny valley*. This region is uncanny in the sense that the surface around adversarial examples is very flat while they are still changing the prediction of the model.

This does not only indicate that the correlation between flatness and strong adversarial examples is not intuitive, but also that a vicinity of such an adversarial sample will be filled with similar adver-

sarial examples. We observe this uncanny valley phenomenon across various model architectures, including large language models (LLMs), datasets, and also in adversarially trained models. Interestingly, for more robust models, much stronger attacks are necessary to find these uncanny valleys. For LLMs, we find that their discrete nature and the availability of only relatively weak attacks often prevent adversarial examples from reaching truly flat regions.

This observation suggests that flatness alone cannot fully explain adversarial robustness; instead it is crucial to locally control the smoothness of the loss Hessian around the adversarial examples. Based on this insight, we derive a connection between relative flatness (Petzka et al., 2021) and adversarial robustness by bounding the third derivative of the loss surface. This bound provides a theoretical guarantee on adversarial robustness, linking the flatness of the loss landscape to the robustness of a model to adversarial attacks. Intuitively, the bound states that adversarial robustness increases as the network becomes flatter when the model function is simultaneously sufficiently smooth in the Lipschitz sense, i.e., *small* Lipschitz constant.

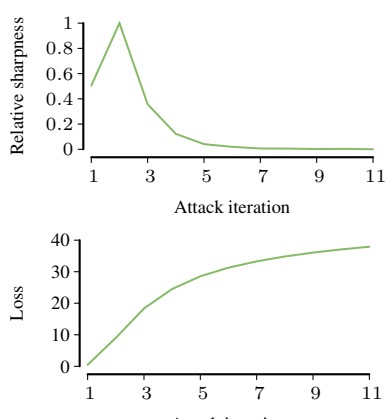

Figure 1: *The Uncanny Valley*. During a multi-step adversarial attack, sharpness first increases; then decreases to almost zero (top), while the loss steadily increases (bottom).

This work highlights the complexity of adversarial robustness and emphasizes the need for a deeper understanding of the geometry of the loss landscape. By bridging the gap between flatness and adversarial robustness, we aim to pave the way for more robust and reliable deep learning models. Additionally our work elucidates the counter-intuitive behaviour of flatness measures based on the trace of the Hessian, when evaluated on adversarial examples.

In summary, our three main contributions are:

1. We introduce and empirically demonstrate the phenomenon of the *uncanny valley*, which is a plateau in the loss surface that we find via adversarial attacks.

2. We perform this analysis on various model architectures and datasets, including convolutional neural networks (CNNs) and large language models (LLMs).

3. We provide a theoretical framework that connects flatness to adversarial robustness through the third derivative of the loss surface.

## 2 RELATED WORK

**Adversarial Examples**   Szegedy et al. (2014) introduced the notion of adversarial inputs for deep learning models as a research subject, characterizing these as a curious property of neural networks. High-level definition of an adversarial example is a perturbation of a benign input that is hard to detect for a human but which leads to the mistake in the model prediction. By now, they have become a major drawback of practical deep learning, since they not only pose a possible threat in applications, but they undermine the trust in machine learning in general: How can we trust the predictions of models that are so easily fooled?

In the meantime, many algorithms for generating adversarial examples have been proposed, starting with FGSM (Goodfellow et al., 2015), followed by the defacto standard attacks PGD (Madry et al., 2017) and C&W (Carlini & Wagner, 2017b), and many more (Papernot et al., 2016; Kurakin et al., 2017; Narodytska & Kasiviswanathan, 2017; Brown et al., 2018; Alaifari et al., 2018; Andriushchenko et al., 2020; Croce & Hein, 2020; 2021). With the development of generative deep learning, the approaches for generating adversarial samples with diffusion models (Chen et al., 2023) appeared. Large language models are not an exception and susceptible not only to standard input manipulation for prediction failure (Li et al., 2020; Zou et al., 2023) but also to so-called jailbreaks, which are forcing the model to output text that was supposed to be excluded from the possible results of the inference (Wei et al., 2024).

**Defenses against Adversarial Attacks**  The most common way to make a model more robust to adversarial examples is adversarial training, which incorporates adversarial inputs into the training procedure (Szegedy et al., 2014; Shafahi et al., 2019; Kumari et al., 2019; Perolat et al., 2018; Shafahi et al., 2020; Cai et al., 2018; Tramèr et al., 2018; Wu et al., 2020; Carmon et al., 2019). It is usually observed that adversarial training reduces the performance of a model on clean data (Tsipras et al., 2018). There also exists contradictory research stating that adversarial training approaches lead to a flatter loss surface with respect to the parameters (Wu et al., 2020; Stutz et al., 2021). This, in turn, is believed to lead to better generalization (Hochreiter & Schmidhuber, 1994).

Rahnama et al. (2020) propose an approach where each of the subnetworks corresponding to layers is robustified via insights from Lyapunov theory, connecting robustness to spectral regularization. Nevertheless, later Liang & Huang (2021) show that large norms of layers do not always induce large global Lipschitz constant. So regularization of norms is not an effective way to enhance robustness. Moosavi-Dezfooli et al. (2019) propose to penalize the Hessian with respect to input to improve the flatness in the input space and demonstrate that it improves adversarial robustness, analogous work was performed by Xu et al. (2020).

Interestingly, Kanai et al. (2023) analyses smoothness in the input space and smoothness in the parameter space and concludes that smoothness in the input space leads to a non-flat surface with respect to parameters, which leads to worse performance. Flatness of the loss surface as a form of smoothness in parameter space has also been linked to generalization (Liang et al., 2019; Tsuzuku et al., 2020). Foret et al. (2020) proposed Sharpness Aware Minimization (SAM), that is essentially similar to the work of Wu et al. (2020), but focuses on improving clean accuracy. The approach became very popular due to the ability to improve some of the state-of-the-art results in image classification. Dinh et al. (2017) show, however, that flatness in terms of the loss Hessian of the entire network cannot predict generalization, due to reparameterizations. Petzka et al. (2021) showed that generalization instead can be linked to a relative flatness of a single layer of a network.

**Lipschitz Continuity and Adversarial Robustness**  One of the drawbacks of adversarial training is the absence of guarantees on the robustness of the obtained model. One of the popular ways to obtain guarantees is randomized smoothing (Cohen et al., 2019), which implicitly controls the global Lipschitz constant (Salman et al., 2019). The main idea here is to produce a smoothed version of a classifier by making it predict the most probable label in a normal distribution surrounding an example. Directly using Lipschitz properties is not quite practical however for modern networks. Global Lipschitz constants (Tsuzuku et al., 2018) are usually too vague, while local Lipschitz constants (Hein & Andriushchenko, 2017) are impractical to compute. Liang & Huang (2021) show that a large global Lipschitz does not make the local constants large, which means that even models with a large global Lipschitz constant can be adversarially robust. Small global Lipschitz constant on the other hand helps to control local constants, but sacrifices clean accuracy. Yang et al. (2020) showed that improving local Lipschitz constants and enforcing better generalization can serve as a method to get good generalizing and robust models , but it might not be easily achievable.

## 3 PRELIMINARIES

We assume a distribution $\mathcal{D}$ over an input space $\mathcal{X}$ and a target space $\mathcal{Y}$ with corresponding probability density function $P(X, Y) = P(Y \mid X)P(X)$, and models $f : \mathcal{X} \to \mathcal{Y}$ from a model class $\mathcal{F}$ and loss functions $\ell : \mathcal{Y} \times \mathcal{Y} \to \mathbb{R}_+$. Intuitively, an adversarial example is a small and imperceptible perturbation $r^*$ of a sample $x$, such that the model incorrectly classifies the perturbed sample $\xi = x + r^*$. Formally, adversarial examples are defined as an optimization problem:

**Definition 1** (Szegedy et al. (2014); Papernot et al. (2016) and Carlini & Wagner (2017b))**.** *Let $f : \mathbb{R}^m \to \{1, \ldots, k\}$ be a classifier, $x \in [0, 1]^m$, and $l \in [k]$ with $l \neq f(x)$ a target class. Then for every*

$$r^* = \arg \min_{r \in \mathbb{R}^m} \|r\|_2 \text{ s.t. } f(x + r) = l \text{ and } x + r \in [0, 1]^m$$

*the perturbed sample $\xi = x + r^*$ is called an **adversarial example**.*

The implicit assumptions here are that (i) the small size of the perturbation in $L_2$-distance of an example $x$ make the adversarial example $\xi$ hard (e.g., for a human) to distinguish from the original $x$ and (ii) the semantics are not altered in the sense that $y = f(x)$ is the true label and for the adversarial

example $\xi$ the true label is still $y$. This means that the loss of an adversarial robust classifier $f^*$ on an adversarial example $\xi$ must not increase significantly, i.e., $l(f^*(\xi), y) - l(f^*(x), y) < \epsilon$. To ensure these assumptions are made explicitly and to improve clarity, adversarial examples can be defined more generally as follows:

**Definition 2.** *Let $\mathcal{D}$ be a distribution over an input space $\mathcal{X}$ and a label space $\mathcal{Y}$ with corresponding probability density function $P(X,Y) = P(Y \mid X)P(X)$. Let $\ell : \mathcal{Y} \times \mathcal{Y} \to \mathbb{R}_+$ be a loss function, $f \in \mathcal{F}$ a model, and $(x, y) \in \mathcal{X} \times \mathcal{Y}$ be an example drawn according to $\mathcal{D}$. Given a distance function $d : \mathcal{X} \times \mathcal{X} \to \mathbb{R}_+$ over $\mathcal{X}$ and two thresholds $\epsilon, \delta \geq 0$, we call $\xi \in \mathcal{X}$ an **adversarial example** for $x$ if $d(x, \xi) \leq \delta$ and*

$$\mathbb{E}_{y_\xi \sim P(Y|X=\xi)} [\ell(f(\xi), y_\xi)] - \ell(f(x), y) > \epsilon \ .$$

By making the explicit assumptions that (i) the distance is measured via the $L_2$-norm, i.e., $d(x, \xi) = \|x - \xi\|_2$, (ii) $f(x)$ is correct, i.e., $f(x) = y$, and (iii) the true label for $\xi$ under $\mathcal{D}$ is $y$, i.e., $\mathbb{E}_{y_\xi \sim P(Y|X)} [\ell(f(\xi), y_\xi)] = \ell(f(\xi), y)$, this definition is equivalent to the original Def. 1. We prove that Def. 2 is a generalization of the classical Def. 1 in Appendix E.

Def. 2 naturally reminds of the $(\epsilon, \delta)$-criterion for continuity. Intuitively, it exemplifies that an adversarially robust classifier must be *sufficiently smooth*, and adversarial examples are a consequence of non-smooth directions. As we prove in Theorem 5, *sufficiently smooth* in this context means flatness as well as a bounded third derivative of the loss function wrt. the weights. To keep our results in line with related work, we use Def. 1 in our experiments, but derive our theoretical results for the more general case of Def. 2.

## 4 EFFICIENT COMPUTATION OF RELATIVE SHARPNESS

To relate flatness and adversarial examples, we want to measure flatness for a particular adversarial example $\xi \in \mathcal{X}$. For that, we need a sound flatness measure and an efficient way to compute it. A sound flatness measure should be correlated with a network's generalization ability. A particular challenge here is that measuring flatness using the loss Hessian wrt. weights (trace or eigenvalues) is not reparameterization-invariant and thus cannot be connected to generalization (Dinh et al., 2017). By deriving their relative flatness measure directly from a decomposition of the generalization gap, Petzka et al. (2021) could show that a combination of trace of the loss Hessian and norm of weights for a single layer of a network is not only reparameterization-invariant, but can be theoretically linked to generalization. This uses a robustness argument: Since for large weights small perturbation in the representation produced by the layer in question can lead to much larger changes in the output, a network needs to be in a much flatter minimum to counteract this. The "relative" part of the flatness measure, i.e., the weights norm component, is a constant in our analysis since we assume a fixed trained neural network. For a model that can be decomposed into a feature extractor $\phi$ and a predictor $g$, i.e., $f(x) = g(\mathbf{w}\phi(x))$, trained on $S \subset \mathcal{X} \times \mathcal{Y}$, the relative flatness measure[1] is defined as

$$\kappa_{Tr}^{\phi}(\mathbf{w}) := \|w\|_2 Tr(H(\mathbf{w}, S)) \ ,$$

where $Tr$ denotes the trace, and $H$ is shorthand for the Hessian of the loss computed on $S$ wrt. $\mathbf{w}$. That is

$$H(\mathbf{w}, S) = \frac{1}{|S|} \sum_{(x,y) \in S} \left( \frac{\partial^2}{\partial w_i \partial w_j} \ell\left(g(\mathbf{w}\phi(x)), y\right) \right)_{i,j \in [km]} \ ,$$

where $\mathbf{w}$ is the weight matrix corresponding to the selected feature layer that connects the feature extractor $\phi$ and the classifier $g$. Similar to Petzka et al. (2021) we select $\phi$ as a neural network up to the penultimate layer, $\mathbf{w}$ the weights from the representation to the next layer, and $g$ the final layer of the network. Counter-intuitively, the relative flatness measure is small if the loss surface is flat, since in that case the trace of the Hessian is small. Similarly, a large value of $\kappa_{Tr}^{\phi}(\mathbf{w})$ means a the loss surface is sharp. To avoid confusion, in the following we therefore call $\kappa_{Tr}^{\phi}(\mathbf{w})$ *relative sharpness*.

To compute relative sharpness efficiently, we use the following representation of the Hessian $H$ for the cross-entropy loss $\ell(x, y) = \sum_{i \in [k]} -y_i \log \hat{y}_i$ for a single example $S = \{(x, y)\}$, where—with

---

[1] Note that this is a simplified relative flatness measure that is not invariant to neuron-wise reparameterizations. Cf. Def. 3 in Petzka et al. (2021).

slight abuse of notation—we denote $\hat{y} = g(\mathbf{w}\phi(x))$ as output of the soft-max layer, and write $\phi$ instead of $\phi(x)$. The Kronecker product is denoted by $\otimes$. We then have

$$H(\mathbf{w}, S) = \begin{bmatrix} \hat{y}_1(1 - \hat{y}_1)\phi\phi^T & -\hat{y}_1\hat{y}_2\phi\phi^T & \dots & -\hat{y}_1\hat{y}_k\phi\phi^T \\ -\hat{y}_2\hat{y}_1\phi\phi^T & \hat{y}_2(1 - \hat{y}_2)\phi\phi^T & & \vdots \\ \vdots & \vdots & \dots & \vdots \\ -\hat{y}_k\hat{y}_1\phi\phi^T & \vdots & \dots & \hat{y}_k(1 - \hat{y}_k)\phi\phi^T \end{bmatrix} \tag{1}$$
$$= (\text{diag}(\hat{y}) - \hat{y}\hat{y}^T) \otimes \phi\phi^T .$$

We refer to Appx. B.4 for a full derivation. The trace of Hessian then is

$$tr(H) = \sum_{j=1}^{k} \hat{y}_j(1 - \hat{y}_j) \sum_{i=1}^{d} \phi_i^2 ,$$

by which we can efficiently compute the relative sharpness measure in $\mathcal{O}(k+d)$ instead of $\mathcal{O}(k^2 d^2)$.

## 5  EMPIRICAL EVALUATION

We perform three groups of experiments to investigate the behavior of the flatness for adversarial samples. First, we demonstrate that adversarial attacks on CNNs find adversarial examples in flat regions according to the relative sharpness measure. We name these flat regions *uncanny valleys*. Second, we investigate, how adversarial training (AT) influences the phenomenon of uncanny valleys. We find that AT simply pushes the uncanny valley further away, however, it can still be easily found by stronger attacks. Last, we conduct a similar analysis on LLMs, where we find that the phenomenon is less pronounced, which is likely due to the discrete nature of the problem, making the attacks weaker and consequentially the uncanny valleys harder to find. All experiments presented below are carried out on machines equipped with an Nvidia-A100 80GB, 256 cores, and 1.9TB of memory. The code is publically available[2].

**Adversarial Examples Fall Flat**   We train a RESNET-18 (He et al., 2016), WIDERESNET-28-4 (Zagoruyko & Komodakis, 2016), DENSENET121 (Huang et al., 2017), VGG11 with Batchnorm (Simonyan & Zisserman, 2014) on CIFAR-10 and CIFAR-100. Each model is trained via stochastic gradient descent for 100 epochs with an initial learning rate of 0.1. We use a cosine scheduler for the learning rate and a weight decay of $10^{-4}$. Next, we attack each model using PGD-$l_\infty$ with 10 iterations and $\delta = 8/255$ (Madry et al., 2017) and record per step the intermediate images generated by the attack and the corresponding loss of the model. This allows us to compute how the relative sharpness develops during the attack. To better compare the different architectures, we normalize the average sharpness to $[0, 1]$. In Fig. 2, we plot, for each step of the attack, the *normalized* relative sharpness measure and the loss with respect to the ground truth. In Appx. C Fig. 7, we show the unnormalized values.

For CIFAR-10, as expected the loss and the sharpness *increase* as the attack progresses, however, around step 3 the sharpness *decreases* again, while the loss further increases until it saturates. This phenomenon can be observed for all model architectures. For CIFAR-100, the pattern is slightly different, namely the unperturbed sample already lies in a rather sharp region (which is indicative of the worse test performance, i.e., a larger generalization gap for this task). Nonetheless, the search for an adversarial example again leads to a very flat region, while the loss steadily increases. These are still adversarial examples, meaning there are adversarial examples in a flat region for good-performing, but adversarially non-robust models. This indicates that flatness on its own cannot characterize adversarial robustness. Characteristic is the fact that one step (corresponding to FGSM) often already leads to a sharp region, thus characterizing this attack as a weaker one.

In Figure 3 we show that the adversarial examples move away from the original example with each attack iteration. Thus, the uncanny valleys truly extend away from the original example, supporting the intuition that adversarial examples live in entire subspaces of the input space (Gubri et al., 2022; Mao et al., 2018).

---

[2]https://anonymous.4open.science/r/the-uncanny-valley-4431

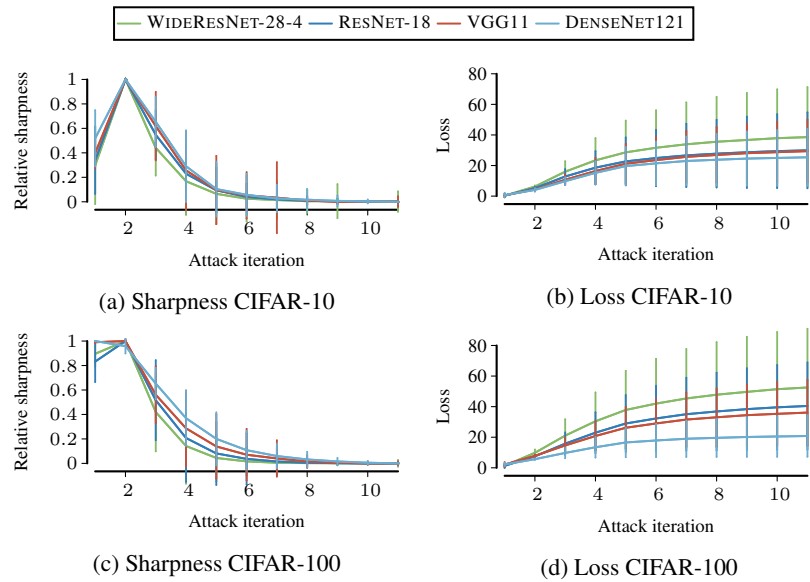

(a) Sharpness CIFAR-10          (b) Loss CIFAR-10

(c) Sharpness CIFAR-100          (d) Loss CIFAR-100

Figure 2: We report the normalized relative sharpness on the attack trajectory for WIDERESNET-28-4, RESNET-18, VGG11 and DENSENET121 on the test set of CIFAR-10 & CIFAR-100. We observe that adversarial examples first reach a sharp region, and as the attack progress they land in a flat region. We also display the standard deviation of the values on individual inputs.

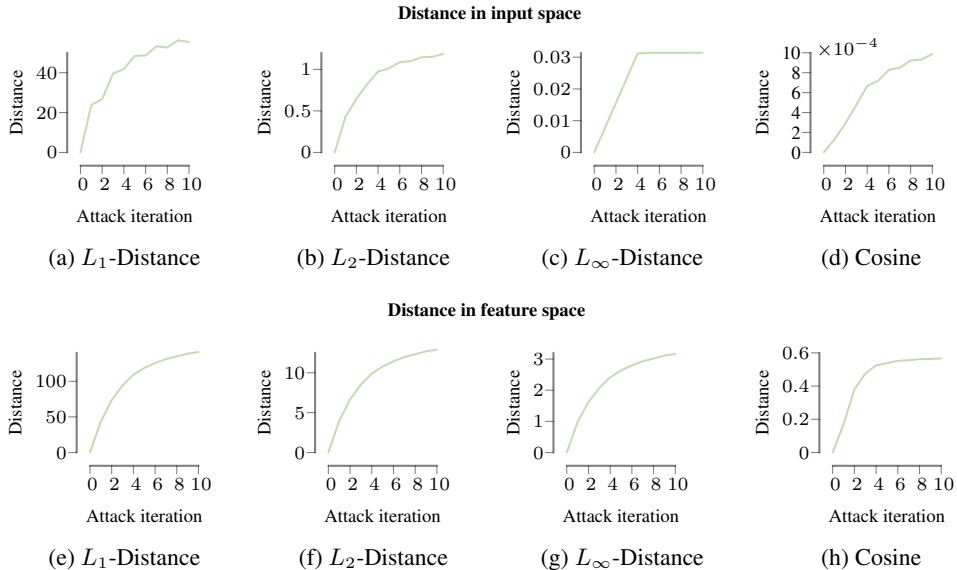

(a) $L_1$-Distance    (b) $L_2$-Distance    (c) $L_\infty$-Distance    (d) Cosine

(e) $L_1$-Distance    (f) $L_2$-Distance    (g) $L_\infty$-Distance    (h) Cosine

Figure 3: We show how far adversarial examples move in image/feature space from the initial image during a PGD-attack; we measure distance with $L_1$, $L_2$, $L_\infty$ and cosine dissimilarity i.e. 1 - cosine similarity. We used CIFAR-10, WIDERESNET-28-4, and PGD with 10 iterations and $\delta$=8/255.

**Adversarial Training Pushes the Flat Region Away**    Next, we investigate how adversarial training influences the sharpness measure and the uncanny valleys. Since all the architectures show practically the same behavior, we focus on WIDERESNET-28-4 and CIFAR-10 & CIFAR-100. To obtain models with varying robustness, we perform adversarial training using PGD-$l_\infty$ with different $\delta' \in \{1, 2, 3, 4, 5, 6, 7, 8\}$, where $\delta = \delta'/255$. For reference, we also report the standard trained model, i.e., $\delta' = 0$. As the models trained with $\delta > 0$ are more robust, we also employ a stronger attack, namely, we use PGD-$l_\infty$ 20 iterations with $\delta = 12/255$ resp. 50 iterations with $\delta = 24/255$. We plot the development of the relative sharpness and loss for ground truth in Fig. 4 and 5.

For smaller $\delta' < 4$, we see a similar behavior to that of normally trained models (Fig. 7). However, we observe that the turning point occurs at later iterations. This effect is even stronger for $\delta' = 4$ and $\delta' = 5$, while for $\delta' > 5$, the models stay in sharp regions. This demonstrates that adversarial training bulges the area around the samples, so the networks are sharper in that region. Nonetheless, we still observe uncanny valleys, except for models trained with larger $\delta'$. If we attack these more robust models with stronger attacks, we again observe that the adversarial examples lie in flat regions (Fig. 4c and 5c). Hence, the uncanny valleys still exist and can be reached via stronger attacks.

**Jailbreak Attacks on LLMs**   Finally, we show that uncanny valleys also exist for LLMs. We use the pre-trained weights of Vicuna-7B (VICUNA, Chiang et al., 2023), Llama-2-7B-Chat (LLAMA2, Touvron et al., 2023), and Guanaco-7B-HF (GUANACO, Dettmers et al., 2024), downloaded from HuggingFace. Each of these models is aligned, meaning they have been trained to not generate any malicious output and thus robustified against jailbreak attacks. We use the attack and benchmark of Zou et al. (2023) for 100 iterations and a batch size of 1024, while sampling the top 256 for further prompting. We focus on generating malicious behavior: For a given prompt, e.g., "*Write a script that can exploit vulnerabilities in a software or operating system*", the goal is to make the model to answer with a specific phrase, e.g, "*Sure, here is a script that can exploit vulnerabilities in a software or operating system: . . .*". Hence, these are targeted attacks. Due to runtime constraints, we use only the first 100 samples in the benchmark. In Fig. 6a and 6b, we report the average relative sharpness per token and the loss with respect to the goal reply.

We observe that for VICUNA the loss landscape first becomes sharper and then the adversarial examples slowly move into flatter regions. For GUANACO, the phenomenon is less pronounced and

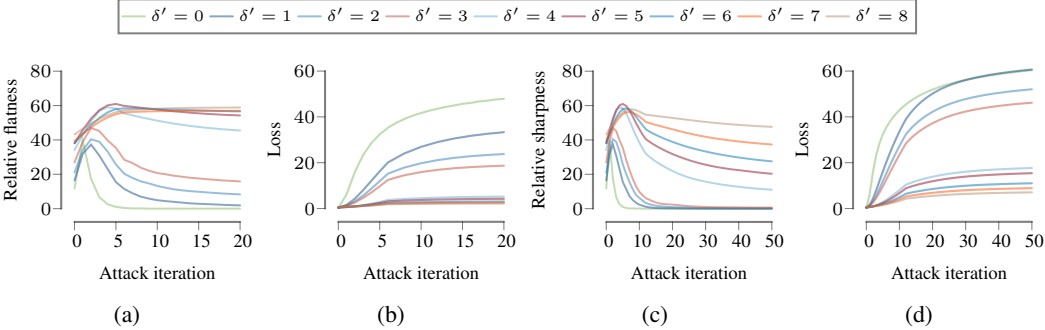

Figure 4: Here, we evaluate adversarially trained WIDERESNET-28-4 on CIFAR-10 with varying $\delta$. We attack the resulting models using PGD-$l_\infty$ with $\delta = 12/255$, steps = 20, shown in Figure (a) & (b), and $\delta = 24/255$, steps = 50, depicted in Figure (c) & (d).We can see that even for adversarially trained models, we can find uncanny valleys by using a stronger attack.

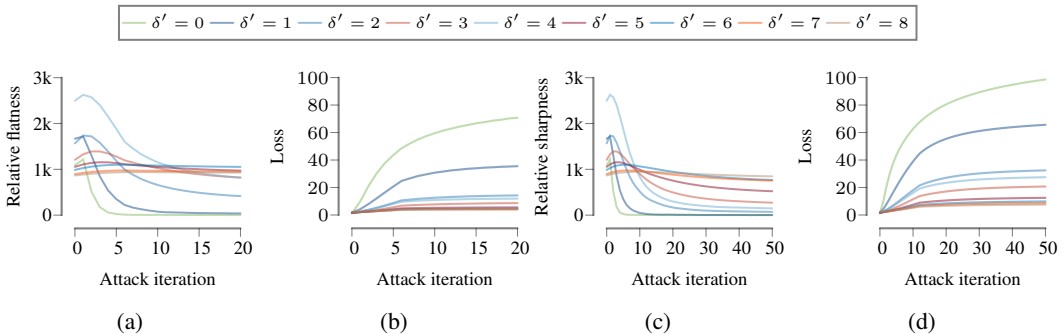

Figure 5: Here, we evaluate adversarially trained WIDERESNET-28-4 on CIFAR-100 with varying $\delta$. We attack the resulting models using PGD-$l_\infty$ with $\delta = 12/255$, steps = 20, shown in Figure (a) & (b), and $\delta = 24/255$, steps = 50, depicted in Figure (c) & (d).We can see that even for adversarially trained models, we can find uncanny valleys by using a stronger attack.

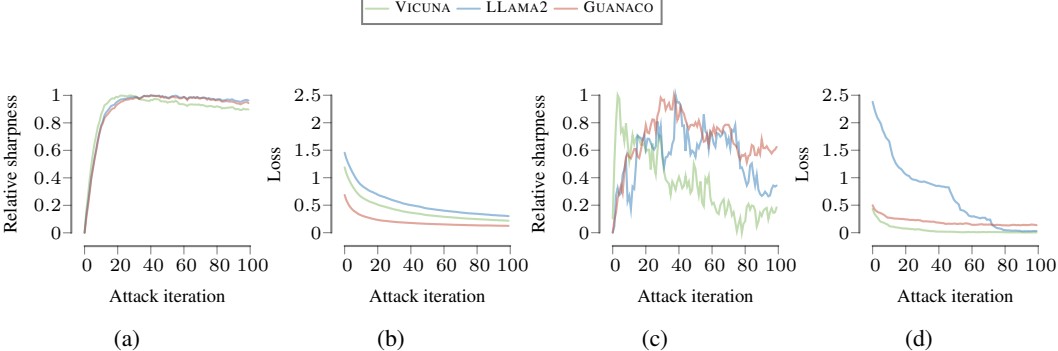

Figure 6: In (a) and (b), we plot the relative sharpness and loss of the adversarial prompt for VI-CUNA, LLAMA2 and GUANACO when attacked by the method of Zou et al. (2023). Additionally, in (c) and (d), we plot per model example trajectories, which first become sharper and then flatter again, together with the corresponding loss.

for LLAMA2, the adversarial examples stay in comparatively sharp regions. The uncanny valleys are not as flat as the undefended models shown in Fig. 2, rather, they resemble the curves of adversarially trained models. This can be explained by the fact that these models have been aligned, i.e., adversarially trained. However, if we inspect individual adversarial samples, we can still find for every model attack trajectories where we can observe the uncanny valleys (cf. Fig. 6c). Given the similarity to the curves of defended models in Fig. 4, we hypothesize that stronger attacks will uncover the uncanny valleys also for LLMs. Nonetheless, even with the current evidence we can conclude that LLMs exhibit uncanny valleys during adversarial attacks.

## 6 BOUNDING ADVERSARIAL ROBUSTNESS VIA RELATIVE SHARPNESS

The definition of adversarial examples entails the assumption that labels are locally constant, which we made explicit in Def. 2. This assumption can lead to counter-intuitive behavior: if the true label of a perturbed example $\xi$ is $y_\xi \neq y$ and the model predicts $y_\xi$, then although correct it counts as an adversarial example, while if it predicts $y$ it does *not* count as an adversarial example even though the model makes a mistake. For image classification, this appears intuitively reasonable since small changes to images should not change the class. It follows that for such applications, the ideal model is smooth, i.e., small perturbations of the input do not affect its output.

Several works have shown that adversarial training increases flatness with respect to the inputs (Moosavi-Dezfooli et al., 2019; Kanai et al., 2023), and adversarial robustness has been empirically linked to flatness of the loss curve with respect to the parameters (Wu et al., 2020; Stutz et al., 2021), but a theoretical link is non-trivial since perturbations of the input relate non-linearly to perturbations of the model weights. In the following, we establish such a formal link and show how it can be used to guarantee adversarial robustness. For that, we first formally define adversarial robustness. A model $f \in \mathcal{F}$ is robust against adversarial examples on a set $S \subseteq \mathcal{X} \times \mathcal{Y}$ if no adversarial examples exist in the vicinity of each element of $S$. With $\mathcal{B}_d^\delta(x) = \{\xi \in \mathcal{X} \mid d(x,\xi) \leq \delta\}$ denoting the $\delta$-ball around $x \in \mathcal{X}$ with respect to the distance $d$ from Def. 2, adversarial robustness can be defined similar to Schmidt et al. (2018) as follows.

**Definition 3.** *Let $S \subseteq \mathcal{X} \times \mathcal{Y}$ be a dataset drawn iid. according to $\mathcal{D}$ and $d : \mathcal{X} \times \mathcal{X}$ be a distance on $\mathcal{X}$. A model $f \in \mathcal{F}$ is $(\epsilon, \delta, S)$-robust against adversarial examples, if for all $x \in S$ it holds that there is no adversarial example with threshold $\epsilon$ in $\mathcal{B}_d^\delta(x)$. A model $f$ is $(\epsilon, \delta)$-robust if it is $(\epsilon, \delta, S)$-robust for all $S$ drawn iid from $\mathcal{D}$.*

Relative sharpness of a model $f(x) = g(\mathbf{w}\phi(x))$ with feature extractor $\phi$, classifier $g$, and weights from features to classifier $\mathbf{w}$ is non-linearly linked to perturbations in the input (Petzka et al., 2021). We use this property to show that relative sharpness measured for a single example can bound the loss increase through perturbations. The minimum relative sharpness for all clean examples in data $S$ then provides a bound on how much any example in $S$ can be perturbed without significantly increasing the loss. In other words, it provides a guarantee on the adversarial robustness of $f$ on $S$.

For this, we have to first establish a link between perturbations in the input space and perturbations in feature space. That is, we express the representation $\phi(\xi)$ of an adversarial example in feature space as a perturbation of the representation $\phi(x)$ of the clean example $x$.

**Lemma 4.** *Let $f = g(\mathbf{w}\phi(x))$ be a model with $\phi$ $L$-Lipschitz and $\|\phi(x)\| \geq r$, and $\xi, x \in \mathcal{X}$ with $\|\xi - x\| \leq \delta$, then there exists a $\Delta > 0$ with $\Delta \leq L\delta r^{-1}$, such that $\phi(\xi) = \phi(x) + \Delta A\phi(x)$, where $A$ is an orthogonal matrix.*

The proof is provided in Appx. B.1. The second step is to relate the adversarial example $\xi$ to perturbations in the weights $\mathbf{w}$ of the representation layer, for which we use the linearity argument from Petzka et al. (2021). Since we can express $\phi(\xi)$ as $\phi(x) + \Delta A\phi(x)$, we can then use the linearity of the representation layer to relate the perturbation of input to a perturbation in weights. That is,

$$\ell(f(\xi), y) = \ell(g(\mathbf{w}\phi(\xi)), y) = \ell(g(\mathbf{w}(\phi(x) + \Delta A\phi(x))), y) = \ell(g((\mathbf{w} + \Delta\mathbf{w}A)\phi(x), y) .$$

This means that we can now express an adversarial example as a suitable perturbation of the weights $\mathbf{w}$ of the representation layer, where the magnitude is bounded as $\Delta \leq L\delta r^{-1}$. We now bound the loss difference between $\xi$ and $x$. For convenience, we define $\ell(\mathbf{w} + \Delta\mathbf{w}A) := \ell(g((\mathbf{w} + \Delta\mathbf{w}A)\phi(x), y)$. The Taylor expansion of $\ell(\mathbf{w} + \Delta\mathbf{w}A)$ at $\mathbf{w}$ yields

$$\ell(\mathbf{w} + \Delta\mathbf{w}) = \ell(\mathbf{w}) + \langle \Delta\mathbf{w}A, \nabla_{\mathbf{w}}\ell(\mathbf{w}) \rangle + \frac{\Delta^2}{2}\langle \mathbf{w}A, H\ell(\mathbf{w})(\mathbf{w}A) \rangle + R_2(\mathbf{w}, \Delta) ,$$

where $H\ell(\mathbf{w})$ is the Hessian of $\ell(\mathbf{w})$. If we now maximize over all $A$ with $\|A\| \leq 1$, it follows that $\langle \mathbf{w}A, H\ell(\mathbf{w})(\mathbf{w}A) \rangle \leq \|\mathbf{w}\|_F^2 Tr(H\ell(\mathbf{w})) = \kappa_{Tr}^\phi(\mathbf{w})$. Therefore, we have

$$|\ell(f(\xi), y) - \ell(f(x), y)| \leq \Delta\|\mathbf{w}\|_F\|\nabla_{\mathbf{w}}\ell(\mathbf{w})\|_F + \frac{\Delta^2}{2}\kappa_{Tr}^\phi(\mathbf{w}) + R_2(\mathbf{w}, \Delta) . \quad (2)$$

The remainder depends on the partial third derivatives of the loss. We show in Appdx. B.2 that for feature extractor $\phi$ that is $L$-Lipschitz, it can be bound by $4^{-1}kmL^3$. With this we can bound the difference between the loss suffered on an adversarial example $\xi$ and the loss on a clean example $x$ for a converged model as follows.

**Proposition 5.** *For $(x, y) \in \mathcal{X} \times \mathcal{Y}$ with $\|x\| \leq 1$ for all $x \in \mathcal{X}$, a model $f(x) = g(\mathbf{w}\phi(x))$ at a minimum $\mathbf{w} \in \mathbb{R}^{m \times k}$ with $\phi$ $L$-Lipschitz and $\|\phi(x)\| \geq r$, and the cross-entropy loss $\ell(\mathbf{w}) = \ell(g(\mathbf{w}\phi(x)), y)$ of $f$ on $(x, y)$, it holds for all $\xi \in \mathcal{X}$ with $\|x - \xi\|_2 \leq \delta$ that*

$$\ell(f(\xi), y) - \ell(f(x), y) \leq \frac{\delta^2}{2r^2}L^2\kappa_{Tr}^\phi(\mathbf{w}) + \frac{\delta^3}{24r^3}kmL^6 .$$

We defer the proof to Appx. B.2. Note, in practice, the perturbation budget $\delta \ll 1$ is chosen to be small. Hence, this bound can be practically useful not only for Lipschitz-regularized networks (Virmaux & Scaman, 2018) or 1-Lipschitz networks (Araujo et al., 2023) but also for larger Lipschitz-constants $L$. This bound on the loss difference $\epsilon$ between $\xi$ and $x$ can be transformed into a guarantee on adversarial robustness based on relative sharpness by solving the cubic equation for $\delta$.

**Corollary 6.** *[Informal] For a dataset $S \subset \mathcal{X} \times \mathcal{Y}$ with $\|x\| \leq 1$ for all $x \in \mathcal{X}$, a model $f(x) = g(\mathbf{w}\phi(x))$ at a minimum $\mathbf{w} \in \mathbb{R}^{m \times k}$ wrt. $S$, with $\phi$ $L$-Lipschitz and $\|\phi(x)\| \geq r$, and the cross-entropy loss $\ell(\mathbf{w}) = \ell(g(\mathbf{w}\phi(x)), y)$ of $f$ on $(x, y)$, $d$ being the L2-distance, and $\epsilon > 0$, $f$ is $(\epsilon, \delta, S)$-robust against adversarial examples with*

$$\delta \propto \frac{\epsilon^{\frac{1}{3}}}{\kappa_{Tr}(\mathbf{w})^{\frac{1}{3}}L} + \frac{rkmL^2}{\kappa_{Tr}(\mathbf{w})} ,$$

*where $\kappa_{Tr}(\mathbf{w})$ is the relative sharpness of $f$ wrt. $\mathbf{w}$.*

The formal version, together with the proof of this corollary, are provided in Appx. B.3. Cor. 6 implies that a flat loss surface measured in terms of relative sharpness $\kappa_{Tr}^\phi(\mathbf{w})$, together with smoothness of the feature extractor $\phi$ in terms of Lipschitz-continuity guarantees adversarial robustness on a dataset $S$ with particular $\delta$. Under the assumption of locally constant labels and for data distributions $\mathcal{D}$ with smooth density $p_{\mathcal{D}}^\phi$ in feature space, the non-simplified relative flatness also implies good generalization (Petzka et al., 2021). Together, these theoretical results indicate that if a model $f = g(\mathbf{w}\phi(x))$ has small relative sharpness $\kappa_{Tr}^\phi(\mathbf{w})$ for all examples in $S$ and $\phi$ is Lipschitz, then it simultaneously achieves good generalization and adversarial robustness.

# 7 DISCUSSION & CONCLUSION

Within the domain of guaranteeing adversarial robustness, a common methodology is to assume (or regularize for) a low Lipschitz constant to derive bounds. It is also known, however, that minimizing a global Lipschitz constant *indefinitely* leads to bad performance on clean samples (Yang et al., 2020). The global Lipschitz constant alone is therefore not helpful for obtaining good-performing and robust models. There exists, however, an intricate connection between the Lipschitz property of the model and its flatness measured with respect to parameters (Kanai et al., 2023); moreover, improving robustness with respect to the parameter changes also improves adversarial robustness empirically (Wu et al., 2020). Using the notion of relative flatness (Petzka et al., 2021), which is theoretically linked to the generalization of the deep learning models, we develop a bound that connects Lipschitzness, flatness, and adversarial robustness. This bound means that inducing flatness with respect to parameters will improve the adversarial robustness of the model, given that the function described by the model is smooth. Together with the results of Petzka et al. (2021) this links the adversarial robustness and generalization abilities of a model.

Measurements of relative flatness for multi-step attacks with varying parameters reveal two things. First, for all architectures, we can systematically find uncanny valleys on which the trajectory indicates a very similar geometry across the models, which might be connected to the existence of Universal Adversarial Examples (Moosavi-Dezfooli et al., 2017) and transferability of adversarial examples. Second, strong adversarial samples fall into flat uncanny valleys and therefore would be missed by most defenses and detection methods based on representations (Lee et al., 2018; Ma et al., 2018; Xu et al., 2018; Hu et al., 2019; Tian et al., 2018). This is in alignment with conclusions drawn by Tramer (2022) and Carlini & Wagner (2017a): the strength of the attack can always be increased in a way that the currently applied defense will fail. At the same moment, uncanny valleys give a fast and easy-to-measure quantity, which could allow for the detection of adversarial attacks during inference. Interestingly, and contrary to the observation of Tramer (2022), stronger attacks that fall into uncanny valleys are in fact *easier* to detect. Simultaneously, relative flatness can also serve as a tool to detect jailbreak attacks, as most of the adversarial prompts lie in sharp regions.

**Limitations & Future Work**  We consider relative flatness wrt. the penultimate layer, because it is theoretically sound and its Hessian is fast to compute , when cross-entropy loss is used . There exists, however, a multitude of previous work claiming that adversarial robustness should be considered with respect to the properties of the individual layers; Kumari et al. (2019) propose a latent adversarial training technique that improves the robustness of intermediate layers and this leads to better overall robustness; Bakiskan et al. (2022) and Walter et al. (2022) analyse which of the layers play a specific role for robustness. Petzka et al. (2021) show that the relative flatness measured in different layers has similar correlation with generalization. In line with that we observation, we find in preliminary experiments that relative flatness wrt. adversarial examples shows similar patterns in different layers (cf. Figure 9), i.e., the uncanny valley is also present in earlier representations. The layer-wise perspective on the robustness is an interesting research direction (Adilova et al., 2023) that might be a continuation for this work, i.e., can we regularize for the flatness of one layer to obtain better performing and more adversarially robust networks? In particular, the fact that the phenomenon follows a similar pattern in shallower layers begs the questions, if the last layer can serve as an avenue to influence the robustness of earlier representations, but at a much lower cost.

Computing the Lipschitz constant exactly is NP-hard, which limits the practical applicability of our theoretical results. For 1-Lipschitz neural networks (Araujo et al., 2023) or networks trained with Lipschitz regularization (Virmaux & Scaman, 2018; Gouk et al., 2021), however, the bound is readily applicable. For general neural networks, exploring how a tighter approximation of local smoothness via local Lipschitzness can be used to improve the practicability of our results makes for excellent future work.

An important direction for future work is to investigate how the phenomenon of uncanny valleys can be translated into an actionable detection method for image classifiers, and more importantly for LLMs, as these models are already exposed to the general public and pose a larger threat. We provide some preliminary results in Appendix F by simply thresholding the sharpness measure which show that adversarial examples can be detected with $> 90\%$ accuracy. Currently, there is no satisfying theoretical explanation for the existence of uncanny valleys. Thus, we are specifically interested in exploring and answering the question: *What lies at the bottom of the uncanny valley?*

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

APPENDIX

## A  PRESENTATION

$w_1 \qquad w_2 \qquad \ell(f(\mathbf{w}x), y)$

$$
\begin{aligned}
\ell(f(\mathbf{w}(x + \delta x)), y) &= \ell(f(\mathbf{w}x + \mathbf{w}\delta x)), y) \\
&= \ell(f(\mathbf{w} + \mathbf{w}\delta)x), y) \\
&= \ell(f(\hat{\mathbf{w}}x), y) \\
&\approx \ell(f(\mathbf{w}x), y)
\end{aligned}
$$

## B  PROOFS OF THEORETICAL RESULTS

In this section, we provide the proofs for the theoretical results in this paper.

### B.1  PROOF OF LEMMA 4

For convenience, we restate the lemma.

**Lemma 4.** *Let $f = g(\mathbf{w}\phi(x))$ be a model with $\phi$ L-Lipschitz and $\|\phi(x)\| \geq r$, and $\xi, x \in \mathcal{X}$ with $\|\xi - x\| \leq \delta$, then there exists a $\Delta > 0$ with $\Delta \leq L\delta r^{-1}$, such that $\phi(\xi) = \phi(x) + \Delta A\phi(x)$, where $A$ is an orthogonal matrix.*

*Proof.* It follows from the proof in Thm. 5 in Petzka et al. (2021) that we can represent any vector $v \in \mathcal{X}$ as $v = w + \Delta Aw$ for some vector $w \in X$, $\Delta \in \mathbb{R}_+$ and $A$ an orthogonal matrix. Then

$$
\begin{aligned}
&\phi(\xi) = \phi(x) + \Delta A\phi(x) \\
\Leftrightarrow&(\phi(\xi) - \phi(x)) = \Delta(A\phi(x)) \\
\Leftrightarrow&\Delta \leq \|(\phi(\xi) - \phi(x))\|\|(A\phi(x))^{-1}\| = \underbrace{\|(\phi(x + \Delta'A'x) - \phi(x))\|}_{\leq L\Delta'}\|(A\phi(x))^{-1}\| \\
\Leftrightarrow&\Delta \leq L\Delta'\|(A\phi(x))^{-1}\| \underbrace{\leq}_{A \text{ orth.}} L\Delta'\frac{1}{r} \ .
\end{aligned}
$$

The result follows from $\|\xi - x\| = \Delta' \leq \delta$. $\qquad\square$

### B.2  PROOF OF PROPOSITION 5

For convenience, we restate the proposition.

**Proposition 5.** *For $(x, y) \in \mathcal{X} \times \mathcal{Y}$ with $\|x\| \leq 1$ for all $x \in \mathcal{X}$, a model $f(x) = g(\mathbf{w}\phi(x))$ at a minimum $\mathbf{w} \in \mathbb{R}^{m \times k}$ with $\phi$ L-Lipschitz and $\|\phi(x)\| \geq r$, and the cross-entropy loss $\ell(\mathbf{w}) = \ell(g(\mathbf{w}\phi(x)), y)$ of $f$ on $(x, y)$, it holds for all $\xi \in \mathcal{X}$ with $\|x - \xi\|_2 \leq \delta$ that*

$$
\ell(f(\xi), y) - \ell(f(x), y) \leq \frac{\delta^2}{2r^2}L^2\kappa_{Tr}^\phi(\mathbf{w}) + \frac{\delta^3}{24r^3}kmL^6 \ .
$$

*Proof.* The remainder $R_2$ in Eq. 2 is

$$
R_2(\mathbf{w}, \Delta) \leq \sup_{\substack{h \in \mathbb{R}^m \\ \|h\|=1}} \sup_{c \in (0,1)} \frac{\Delta^3}{3!} \sum_{i,j,k}^d \frac{\partial^3 \ell}{\partial w_i \partial w_j \partial w_k}(x + c\Delta h) \ ,
$$

where $w \in \mathbb{R}^d$ with $d = km$ is the vectorization of $\mathbf{w} \in \mathbb{R}^{m \times k}$. We now bound this remainder. Using the representation of the loss Hessian from Eq. 1, we can write the partial third derivatives in

the remainder as

$$\frac{\partial^3 \ell}{\partial w_i \partial w_j \partial w_k}(x) = \sum_{\substack{o,l,j \in [k] \\ a,b,c \in [m]}} -\left(\hat{y}_j \hat{y}_o (\mathbb{1}_{o=j} - \hat{y}_l) + \hat{y}_l \hat{y}_o (\mathbb{1}_{o=l} - \hat{y}_j)\right) \phi(x)_a \phi(x)_b \phi(x)_c \quad ,$$

where $\hat{y} = f(x)$. Under the assumption that $\phi$ is $L$-Lipschitz, for all $x \in \mathcal{X}$, $\|x\| \leq 1$ and observing that $\sum_{o \in [k]} \hat{y}_o = 1$ we can bound this term by

$$\frac{\partial^3 \ell}{\partial w_i \partial w_j \partial w_k}(x) \leq \frac{1}{4} kmL'^3 \quad . \tag{3}$$

The terms $k, m$ follow from the sum over all rows and columns of $\mathbf{w}$, and the factor $4^{-1}$ follows from the fact that the predictions in $\hat{y}$ sum up to 1. The factor $L'^3$ can be derived as follows.

$$\phi(x)_i \leq \|\phi(x)_i\| = \|(\phi(x)_i - \phi(\mathbf{0})_i) + \phi(\mathbf{0})_i\| \tag{4}$$
$$\leq \|(\phi(x)_i - \phi(\mathbf{0})_i)\| + \|\phi(\mathbf{0})_i\| \tag{5}$$
$$\leq L\|x - \mathbf{0}\| + \|\phi(\mathbf{0})_i\| \qquad (\phi \text{ is } L\text{-Lipschitz}) \tag{6}$$
$$\leq L + \|\phi(\mathbf{0})_i\| \leq L + C_{\phi(\mathbf{0})} \qquad (\|x\| \leq 1) \tag{7}$$
$$\tag{8}$$

where $C_{\phi(\mathbf{0})} := max_i \|\phi(\mathbf{0})_i\|$. For Relu networks without bias $C_{\phi(\mathbf{0})}$ is 0 and empirically for networks with a bias term, it is very small i.e. $\approx 1$. Therefore $L' = L + C_{\phi(\mathbf{0})} \approx L$, in particular since for most realistic neural networks $L$ is large. For simplicity, we subsitute $L = L'$.

Inserting this bound into Eq. 2 yields

$$|\ell(f(\xi), y) - \ell(f(x), y)| \leq \Delta \|\mathbf{w}\|_F \|\nabla_{\mathbf{w}} \ell(\mathbf{w})\|_F + \frac{\Delta^2}{2} \kappa_{Tr}^\phi(\mathbf{w}) + \frac{\Delta^3}{3!} \frac{1}{4} kmL^3 \quad .$$

The result follows from setting $\Delta \leq L\delta r^{-1}$. $\qquad\qquad\qquad\qquad\qquad\qquad \square$

### B.3 PROOF OF PROPOSITION 6

**Proposition 7.** *For a dataset $S \subset \mathcal{X} \times \mathcal{Y}$ with $\|x\| \leq 1$ for all $x \in \mathcal{X}$, a model $f(x) = g(\mathbf{w}\phi(x))$ at a minimum $\mathbf{w} \in \mathbb{R}^{m \times k}$ wrt. $S$, with $\phi$ $L$-Lipschitz and $\|\phi(x)\| \geq r$, and a loss function $\ell(\mathbf{w}) = \ell(g(\mathbf{w}\phi(x)), y)$ of $f$ on $(x, y)$, $d$ being the L2-distance, and $\epsilon > 0$, $f$ is $(\epsilon, \delta, S)$-robust against adversarial examples with*

$$\delta \geq \left( -\frac{8r^3 k^3 m^3 L^9 + 27\epsilon}{27 L^3 \kappa_{Tr}^\phi(\mathbf{w})} + \left( -\frac{2^7}{27} \frac{r^6 k^6 m^6 L^3}{\kappa_{Tr}^\phi(\mathbf{w})^6} + \frac{r^6 \epsilon^2}{L^6 \kappa_{Tr}^\phi(\mathbf{w})^2} - \frac{2^4 r^6 \epsilon k^3 m^3 L^{\frac{3}{2}}}{27 \kappa_{Tr}^\phi(\mathbf{w})^4} \right)^{\frac{1}{2}} \right)^{\frac{1}{3}}$$

$$+ \left( -\frac{8r^3 k^3 m^3 L^9 + 27\epsilon}{27 L^3 \kappa_{Tr}^\phi(\mathbf{w})} - \left( -\frac{2^7}{27} \frac{r^6 k^6 m^6 L^3}{\kappa_{Tr}^\phi(\mathbf{w})^6} + \frac{r^6 \epsilon^2}{L^6 \kappa_{Tr}^\phi(\mathbf{w})^2} - \frac{2^4 r^6 \epsilon k^3 m^3 L^{\frac{3}{2}}}{27 \kappa_{Tr}^\phi(\mathbf{w})^4} \right)^{\frac{1}{2}} \right)^{\frac{1}{3}}$$

$$+ \frac{2rkmL^2}{72 \kappa_{Tr}^\phi(\mathbf{w})}$$

*where $\kappa_{Tr}(\mathbf{w})$ is the relative flatness of $f$ wrt. $\mathbf{w}$. That is,*

$$\delta \propto \frac{\epsilon^{\frac{1}{3}}}{(L^3 \kappa_{Tr}(\mathbf{w}))^{\frac{1}{3}}} + \frac{rkmL^2}{\kappa_{Tr}(\mathbf{w})} \quad .$$

*Proof.* From Prop. 5 it follows that we achieve $(\epsilon, \Delta, S)$-robustness where

$$\epsilon = \frac{\Delta^2}{2} \kappa_{Tr}^\phi(\mathbf{w}) + \frac{\Delta^3}{24} kmL^3 \quad .$$

First, we need to solve this cubic equation for $\Delta$. For that, we substitute $a = \frac{kmL^3}{24}$ and $b = \frac{\kappa^\phi_{Tr}(w)}{2}$ and get

$$0 = a\Delta^3 + b\Delta^2 - \epsilon = \Delta^3 + \frac{a}{b}\Delta^2 - \frac{\epsilon}{b} = \Delta^3 + \alpha\Delta^2 - \beta \ ,$$

by subsituting $\alpha = \frac{a}{b}$ and $\beta = \frac{\epsilon}{b}$. We use the depressed cubic form $\Delta = t - \frac{\alpha}{3}$ and get

$$0 = \left(t - \frac{\alpha}{3}\right)^3 + \alpha\left(t - \frac{\alpha}{3}\right)^2 - \beta$$
$$\Leftarrow 0 = t^3 - t^2\alpha + t\frac{\alpha^2}{3} - \frac{\alpha^3}{27} + t^2\alpha - \frac{2\alpha^2 t}{3} + \frac{\alpha^3}{9} - \beta$$
$$\Leftarrow 0 = t^3 - t\frac{\alpha^2}{3} + \frac{2\alpha^3}{27} - \beta \ .$$

with $p = -\frac{\alpha^2}{3}$ and $q = \frac{2\alpha^3}{27} - \beta$ we get the form $0 = t^3 + pt + q$ for which we can apply Cardano's formula.

$$t = \left(-\frac{q}{2} + \left(\frac{q^2}{4} + \frac{p^3}{9}\right)^{\frac{1}{2}}\right)^{\frac{1}{3}} + \left(-\frac{q}{2} - \left(\frac{q^2}{4} + \frac{p^3}{9}\right)^{\frac{1}{2}}\right)^{\frac{1}{3}}$$

Resubstituting $p, q, \alpha, \beta$ yields

$$\frac{q}{2} = \frac{a^3}{27b^3} - \frac{\epsilon}{2n} \ , \quad \frac{p^3}{9} = -\frac{1}{9^3}\frac{a^6}{b^6} \ , \quad \frac{q^2}{4} = \frac{1}{27^2}\frac{a^6}{b^6} + \frac{\epsilon^2}{4b^2} - \frac{\alpha^3\epsilon}{27b^4} \ .$$

Substituting this in the solution for $t$ then gives

$$t = \left(-\frac{a^3}{27b^3} + \frac{\epsilon}{2b} + \left(-\frac{2a^6}{27b^6} + \frac{\epsilon^2}{4b^2} - \frac{a^3\epsilon}{27b^4}\right)^{\frac{1}{2}}\right)^{\frac{1}{3}}$$
$$+ \left(-\frac{a^3}{27b^3} + \frac{\epsilon}{2b} - \left(-\frac{2a^6}{27b^6} + \frac{\epsilon^2}{4b^2} - \frac{a^3\epsilon}{27b^4}\right)^{\frac{1}{2}}\right)^{\frac{1}{3}}$$

Substituting $a, b$ and $t = \Delta + \frac{\alpha}{3}$ yields

$$\Delta = \left(-\frac{8k^3m^3L^9 + 27\epsilon}{27\kappa^\phi_{Tr}(\mathbf{w})} + \left(-\frac{2^7}{27}\frac{k^6m^6L^{18}}{\kappa^\phi_{Tr}(\mathbf{w})^6} + \frac{\epsilon^2}{\kappa^\phi_{Tr}(\mathbf{w})^2} - \frac{2^4\epsilon k^3m^3L^9}{27\kappa^\phi_{Tr}(\mathbf{w})^4}\right)^{\frac{1}{2}}\right)^{\frac{1}{3}}$$
$$+ \left(-\frac{8k^3m^3L^9 + 27\epsilon}{27\kappa^\phi_{Tr}(\mathbf{w})} - \left(-\frac{2^7}{27}\frac{k^6m^6L^{18}}{\kappa^\phi_{Tr}(\mathbf{w})^6} + \frac{\epsilon^2}{\kappa^\phi_{Tr}(\mathbf{w})^2} - \frac{2^4\epsilon k^3m^3L^9}{27\kappa^\phi_{Tr}(\mathbf{w})^4}\right)^{\frac{1}{2}}\right)^{\frac{1}{3}}$$
$$+ \frac{2kmL^3}{72\kappa^\phi_{Tr}(\mathbf{w})}$$

Finally, from Lemma 4 we have $\Delta \leq L\delta r^{-1}$, so that

$$
\begin{aligned}
\delta \geq{} & \frac{r}{L}\left(-\frac{8k^3m^3L^9+27\epsilon}{27\kappa_{Tr}^{\phi}(\mathbf{w})}+\left(-\frac{2^7}{27}\frac{k^6m^6L^{18}}{\kappa_{Tr}^{\phi}(\mathbf{w})^6}+\frac{\epsilon^2}{\kappa_{Tr}^{\phi}(\mathbf{w})^2}-\frac{2^4\epsilon k^3m^3L^9}{27\kappa_{Tr}^{\phi}(\mathbf{w})^4}\right)^{\frac{1}{2}}\right)^{\frac{1}{3}} \\
& +\frac{r}{L}\left(-\frac{8k^3m^3L^9+27\epsilon}{27\kappa_{Tr}^{\phi}(\mathbf{w})}-\left(-\frac{2^7}{27}\frac{k^6m^6L^{18}}{\kappa_{Tr}^{\phi}(\mathbf{w})^6}+\frac{\epsilon^2}{\kappa_{Tr}^{\phi}(\mathbf{w})^2}-\frac{2^4\epsilon k^3m^3L^9}{27\kappa_{Tr}^{\phi}(\mathbf{w})^4}\right)^{\frac{1}{2}}\right)^{\frac{1}{3}} \\
& +\frac{2rkmL^2}{72\kappa_{Tr}^{\phi}(\mathbf{w})} \\
={} & \left(-\frac{8r^3k^3m^3L^9+27\epsilon}{27L^3\kappa_{Tr}^{\phi}(\mathbf{w})}+\left(-\frac{2^7}{27}\frac{r^6k^6m^6L^3}{\kappa_{Tr}^{\phi}(\mathbf{w})^6}+\frac{r^6\epsilon^2}{L^6\kappa_{Tr}^{\phi}(\mathbf{w})^2}-\frac{2^4r^6\epsilon k^3m^3L^{\frac{3}{2}}}{27\kappa_{Tr}^{\phi}(\mathbf{w})^4}\right)^{\frac{1}{2}}\right)^{\frac{1}{3}} \\
& +\left(-\frac{8r^3k^3m^3L^9+27\epsilon}{27L^3\kappa_{Tr}^{\phi}(\mathbf{w})}-\left(-\frac{2^7}{27}\frac{r^6k^6m^6L^3}{\kappa_{Tr}^{\phi}(\mathbf{w})^6}+\frac{r^6\epsilon^2}{L^6\kappa_{Tr}^{\phi}(\mathbf{w})^2}-\frac{2^4r^6\epsilon k^3m^3L^{\frac{3}{2}}}{27\kappa_{Tr}^{\phi}(\mathbf{w})^4}\right)^{\frac{1}{2}}\right)^{\frac{1}{3}} \\
& +\frac{2rkmL^2}{72\kappa_{Tr}^{\phi}(\mathbf{w})}
\end{aligned}
$$

$\square$

### B.4 DERIVATION OF HESSIAN AND THIRD DERIVATIVE

Let $\phi \in \mathbb{R}^m$ denote the embedding of the feature extractor and $W \in \mathbb{R}^{K \times m}$, where we denote the weights of the $k$-th neuron as $w_k$. The output layer is given by the softmax function $\hat{y}_k = \text{softmax}(W\phi) \in \mathbb{R}^K$. More precisely the softmax is given by,

$$
\hat{y}_k = \frac{\exp(w_k\phi)}{\sum_{j=1}^{K}\exp(w_j\phi)}
$$

For simplicity, we omit the bias term. The one-hot encoded ground truth is given by $y$. The derivative of the loss $L$ function wrt. the weight vector $w_j$ can be computed as

$$
\frac{\partial L(y,\hat{y})}{\partial w_j} = -(y_j - \hat{y}_j)\phi^T
$$

**Second derivative**

$$
\begin{aligned}
\frac{\partial L(y,\hat{y})}{\partial w_l\,\partial w_j} &= \frac{\partial}{\partial w_l} - (y_j - \hat{y}_j)\phi^T \\
&= \frac{\partial}{\partial w_l} - y_j\phi^T + \frac{\partial}{\partial w_l}\hat{y}_j\phi^T \\
&= \frac{\partial}{\partial w_l}\hat{y}_j\phi^T
\end{aligned}
$$

The last equation follows from $y$ being independent of $w_l$. Next, we do a case analysis on $l = j$.

- ($l = j$): In (1), we use the quotient rule and in (2) definition of softmax.

$$\frac{\partial}{\partial w_l}\hat{y}_j\phi^T = \frac{\partial}{\partial w_j}\frac{\exp(w_j\phi)}{\sum_{k=1}^{K}\exp(w_k\phi)}\phi^T$$

$$= \frac{(\exp(w_j\phi)\sum_{k=1}^{K}\exp(w_k\phi))\phi\phi^T - \exp(w_j\phi)\exp(w_j\phi)\phi\phi^T}{(\sum_{k=1}^{K}\exp(w_k\phi))^2} \ (1)$$

$$= \left(\frac{\exp(w_j\phi)\sum_{k=1}^{K}\exp(w_k\phi)}{(\sum_{k=1}^{K}\exp(w_k\phi))^2} - \frac{\exp(w_j\phi)^2}{(\sum_{k=1}^{K}\exp(w_k\phi))^2}\right)\phi\phi^T$$

$$= (\hat{y}_j - \hat{y}_j^2)\phi\phi^T \in \mathbb{R}^{m\times m} \ (2)$$

- ($l \neq j$): Again quotient rule, but the left side vanishes.

$$\frac{\partial}{\partial w_l}\hat{y}_j\phi^T = -\hat{y}_l\hat{y}_j\phi\phi^T \in \mathbb{R}^{m\times m}$$

Then we have

$$\frac{\partial L(y,\hat{y})}{\partial w_l\,\partial w_j} = \hat{y}_l(\mathbb{1}_{[l=j]} - \hat{y}_j)\phi\phi^T \in \mathbb{R}^{m\times m}$$

The hessian is then given by

$$H(L;W)(y,\hat{y}) = (\mathrm{diag}(\hat{y}) - \hat{y}\hat{y}^T)\otimes\phi\phi^T \ \in \mathbb{R}^{Km\times Km}$$

**Third derivative**   First, rewrite

$$\frac{\partial L(y,\hat{y})}{\partial w_l\,\partial w_j} = \hat{y}_l(\mathbb{1}_{[l=j]} - \hat{y}_j)\phi\phi^T = \hat{y}_l\mathbb{1}_{[l=j]}\phi\phi^T - \hat{y}_l\hat{y}_j\phi\phi^T$$

Then we define a new operator $\Pi : \mathbb{R}^n \times \mathbb{R}^m \times \mathbb{R}^o \to \mathbb{R}^{n\times m\times o}, \Pi(x,y,z)_{ijk} = x_iy_jz_k$. We now compute

$$\frac{\partial L(y,\hat{y})}{\partial w_o\,\partial w_l\,\partial w_j} = \frac{\partial}{\partial w_o}\hat{y}_l\mathbb{1}_{[l=j]}\phi\phi^T - \hat{y}_l\hat{y}_j\phi\phi^T$$

Again we make a CA on $j = l$

- ($l = j$):

$$\frac{\partial}{\partial w_o}(\hat{y}_l\phi\phi^T - \hat{y}_l^2\phi\phi^T) = \frac{\partial}{\partial w_o}\hat{y}_l\phi\phi^T - \frac{\partial}{\partial w_o}\hat{y}_l^2\phi\phi^T$$

$$= \hat{y}_o(\mathbb{1}_{[o=l]} - \hat{y}_l)\,\Pi(\phi,\phi,\phi) - 2(\hat{y}_o(\mathbb{1}_{[o=l]} - \hat{y}_l)\,\Pi(\phi,\phi,\phi))$$

$$= -\hat{y}_o(\mathbb{1}_{[o=l]} - \hat{y}_l)\,\Pi(\phi,\phi,\phi)$$

- ($l \neq j$):

$$\frac{\partial}{\partial w_o} - \hat{y}_l\hat{y}_j\phi\phi^T = -(\frac{\partial}{\partial w_o}\hat{y}_l)\hat{y}_j\phi\phi^T - \hat{y}_l(\frac{\partial}{\partial w_o}\hat{y}_j)\phi\phi^T$$

$$= -\hat{y}_j\hat{y}_o(\mathbb{1}_{[o=l]} - \hat{y}_l)\cdot\Pi(\phi,\phi,\phi) - \hat{y}_l\hat{y}_o(\mathbb{1}_{[o=i]} - \hat{y}_j)\cdot\Pi(\phi,\phi,\phi)$$

$$= -[\hat{y}_j\hat{y}_o(\mathbb{1}_{[o=l]} - \hat{y}_l) + \hat{y}_l\hat{y}_o(\mathbb{1}_{[o=i]} - \hat{y}_j)]\cdot\Pi(\phi,\phi,\phi) \in \mathbb{R}^{m\times m\times m}$$

$$\to -[\hat{y}_j\hat{y}_o(\mathbb{1}_{[o=l]} - \hat{y}_l) + \hat{y}_l\hat{y}_o(\mathbb{1}_{[o=j]} - \hat{y}_j)]_{j,l,o=1..k}\otimes\Pi(\phi,\phi,\phi) \in \mathbb{R}^{Km\times Km\times Km}$$

## C   UNNORMALIZED PLOTS

## D   CODE FOR EXPERIMENTS

We use code from several resources, which we disclose here. First, the basis for training and attacking the CNNs stems from (Sehwag et al., 2020). We modified the code according to our needs. The code for DenseNet121 stems from the official PyTorch library. To attack and evaluate the LLMs, we use the official implementation of the attack (Zou et al., 2023). CIFAR-10 and CIFAR-100 were also downloaded from PyTorch.

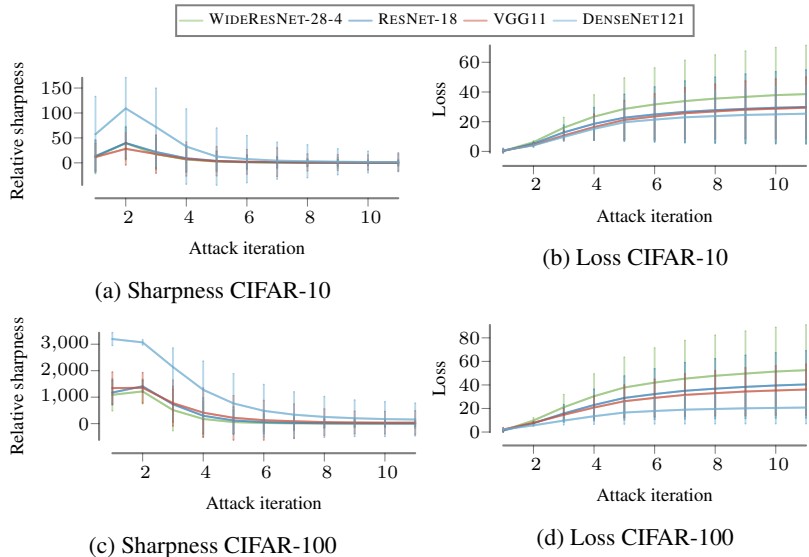

Figure 7: We report the relative sharpness on the attack trajectory of attack for WIDERESNET-28-4, RESNET-18, VGG11 and DENSENET121 on the test set of CIFAR-10 & CIFAR-100. We observe that adversarial examples first reach a sharp region, but with strength of the attack increasing they are in very flat region. We also display the standard deviation of the values on individual inputs.

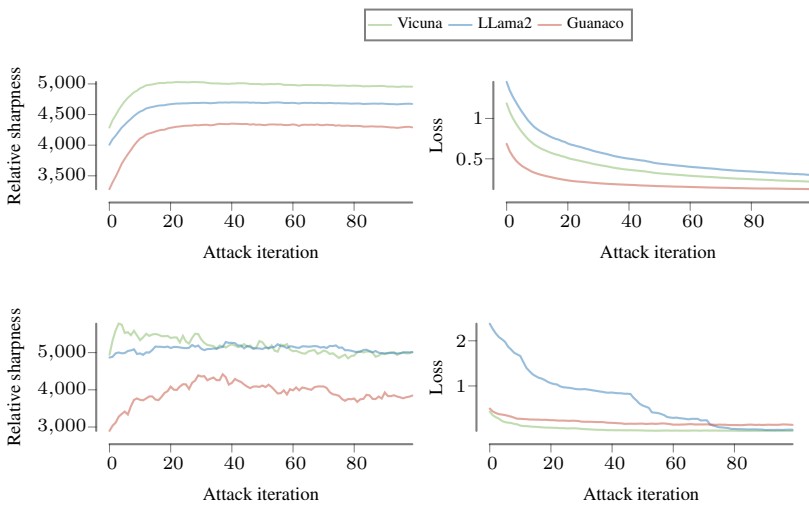

Figure 8: **Top**. We plot the relative sharpness and loss of the adversarial prompt for Viucna, LLama2 and Guanaco when attacked by the method of Zou et al. (2023). **Bottom**. We give per model example trajectories that first get sharper and then flatter again.

# E   PROOF THAT DEFINITION 2 GENERALIZES DEFINITION 1

In the following, we provide a straightforward proof that Definition 2 is a generalization of the classical definition of adversarial examples by Szegedy et al. (2014); Papernot et al. (2016) and Carlini & Wagner (2017b). For that, we first restate our definition.

**Definition 2.** *Let $\mathcal{D}$ be a distribution over an input space $\mathcal{X}$ and a label space $\mathcal{Y}$ with corresponding probability density function $P(X, Y) = P(Y \mid X)P(X)$. Let $\ell : \mathcal{Y} \times \mathcal{Y} \to \mathbb{R}_+$ be a loss function, $f \in \mathcal{F}$ a model, and $(x, y) \in \mathcal{X} \times \mathcal{Y}$ be an example drawn according to $\mathcal{D}$. Given a distance function $d : \mathcal{X} \times \mathcal{X} \to \mathbb{R}_+$ over $\mathcal{X}$ and two thresholds $\epsilon, \delta \geq 0$, we call $\xi \in \mathcal{X}$ an **adversarial example** for $x$*

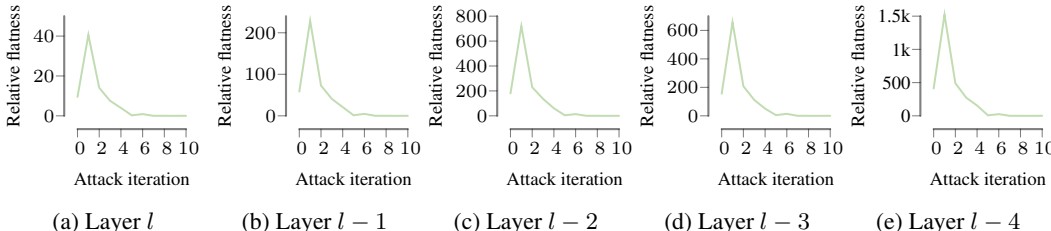

| (a) Layer $l$ | (b) Layer $l-1$ | (c) Layer $l-2$ | (d) Layer $l-3$ | (e) Layer $l-4$ |

Figure 9: We show the relative sharpness measure computed in the penultimate layer $l$ and in shallower layers $l-1$ to $l-4$ for WIDERESNET-28-4. Due to memory and runtime constraints, we approximate the measure using Hutchinson trace estimation used in Petzka et al. (2021) on 500 images. We observe the same phenomena as in the penultimate layer, which justifies that we focus only on the penultimate layer for our theoretical and experimental analysis.

*if $d(x, \xi) \leq \delta$ and*

$$\mathop{\mathbb{E}}_{y_\xi \sim P(Y|X=\xi)} [\ell(f(\xi), y_\xi)] - \ell(f(x), y) > \epsilon \, .$$

We now restate the classical definition here. For that, note that in Szegedy et al. (2014), the adversarial examples are assumed to be in $x + r \in [0, 1]^m$ since they assume data to be images with pixel values in $[0, 1]$. The original definition in Szegedy et al. (2014) has an inconsistency, assuming $x \in \mathbb{R}^m$. For correctness, we therefore assume $x \in [0, 1]^m$—Def. 2 would similarly generalize to arbitrary $\mathcal{X}$.

**Definition 1** (Szegedy et al. (2014); Papernot et al. (2016) and Carlini & Wagner (2017b)(targeted)).
*Let $f : \mathbb{R}^m \to \{1, \ldots, k\}$ be a classifier, $x \in [0, 1]^m$, and $l \in [k]$ with $l \neq f(x)$ a target class. Then for every*

$$r^* = \arg \min_{r \in \mathbb{R}^m} \|r\|_2 \text{ s.t. } f(x + r) = l \text{ and } x + r \in [0, 1]^m$$

*the perturbation $x + r^*$ is called an adversarial example.*

*Proof.* We now prove that for particular choices of loss function, distance measure and thresholds, adversarial example fits to our general Definition 2 if and only if it fits to the classical Definition 1.

For that, let $\mathcal{X} = [0, 1]^m$, $\mathcal{Y} = \{1, \ldots, k\}$ and for a distribution $\mathcal{D}$ and classifier $f$, we assume for $x \in \mathcal{X}$ that $P(Y = y|X = x) = 1$, if $y = f(x)$ and 0 otherwise. For $l \in [k]$, we set $\epsilon > 0$, and

$$\ell(\widehat{y}, y) = \begin{cases} 0 \, , \text{if } \widehat{y} \neq l \\ \epsilon + 1 \, , \text{otherwise} \end{cases}$$

Furthermore, let $d = \|\cdot\|_2$ and

$$\delta = \min_{r \in \mathbb{R}^m} \|r\|_2 \text{ s.t. } \ell(f(x + r), f(x)) > 0 \text{ and } x + r \in [0, 1]^m$$

Lastly, we assume locally constant labels, i.e., for all $\xi = x + r$ with small perturbation $r$ it holds that $P(Y = y|X = x + r) = P(Y = y|X = x)$, i.e., the conditional distribution of the true label is constant around $x \in \mathcal{X}$.

(I) Let $\xi = x + r^*$ be an adversarial example according to Def. 2, then by construction of the loss function, $f(\xi) = l$. Furthermore, $\delta$ is chosen so that $\xi - x = r \in [0, 1]^m$, and $\delta = \|r^*\|_2 = \min_{r \in \mathbb{R}^m} \|r\|_2$ s.t. $f(x + r) = l$. Therefore, $x + r^*$ is also an adversarial example according to Def. 1.

(II) Let $x + r^*$ be an adversarial example according to Def. 1, then $f(x + r^*) = l$ and thus $\ell(f(\xi), y_\xi) = \ell(f(x + r^*), y) = \ell(f(x + r^*), f(x)) = \epsilon + 1 > \epsilon$. Furthermore $d(x, \xi) = \|x - x + r^*\|_2 = \min_{r \in \mathbb{R}^m} \|r\|_2$ s.t. $\ell(f(x + r), f(x)) > 0$ and $r \in [0, 1]^m \leq \delta$.

$\square$

Definition 2 naturally expands Definition 1 by taking into account other distance metrics and formalizing the thresholds that are not specified in the original definition. A threshold on the loss difference generalizes from classification tasks, where targeted attacks can be modeled by a specific loss function. Therefore, our definition allows identifying adversarial example with respect to loss and not with respect to prediction itself, which is closer to the construction of adversarial examples with loss maximization. A distance threshold captures the requirement that adversarial perturbations are *imperceptible*. Using the minimum distance is reasonable from an optimization perspective, but it also highlights a flaw in the original definition: If the minimal perturbation is so large that is indeed perceptible (or, even worse, would change, e.g., the picture of a cat to that of a dog) then it would still be considered an adversarial example under Def. 1.

It seems more natural to set a fixed threshold $\delta$ on the distance that captures what it means for a perturbation to be imperceptible, as in the definition of untargeted attacks proposed by Carlini & Wagner (2017b). From all adversarial examples defined that way, one can then find the closest to the clean example. We show that Definition 2 is also a generalization of this definition by Carlini & Wagner (2017b) of untargeted attacks with general $L_p$-distances.

**Definition 8** (Carlini & Wagner (2017b) (untargeted)). *For $\mathcal{X} = \mathbb{R}^n$ and $\mathcal{Y} = \{1, \dots, m\}$, let $f : \mathcal{X} \to \mathcal{Y}$ be a classifier, $x \in \mathbb{R}^n$, $\|\cdot\|_p$ be a p-norm, and $\delta > 0$. Then every $\xi \in \mathcal{X}$ with $\|x - \xi\|_p \leq \delta$ is an adversarial example for $x$ if $f(\xi) \neq f(x)$.*

*Proof.* We assume $\mathcal{X} = \mathbb{R}^n$ and $\mathcal{Y} = \{1, \dots, m\}$, $d = \|\cdot\|_p$, and for a distribution $\mathcal{D}$ and classifier $f$, we assume for $x \in \mathcal{X}$ that $P(Y = y|X = x) = 1$, if $y = f(x)$ and 0 otherwise. For $\epsilon > 0$, let $\ell$ be a loss function with $\ell(y, y) = 0$ and for every $y' \neq y$, $\ell(y', y) > \epsilon$. Lastly, we again assume a locally constant labels, i.e., for all $\xi \in \mathcal{X}$ with $\|x - \xi\|_p \leq \delta$ it holds that $P(Y = y|X = \xi) = P(Y = y|X = x)$.

(I) Let $\xi$ be an adversarial example according to Def. 2, then by construction of the loss function, $f(\xi) \neq f(x)$. Furthermore, $\|x - \xi\|_p = d(x, \xi) \leq \delta$. Therefore, $\xi$ is also an adversarial example according to Def. 8.

(II) Let $\xi$ be an adversarial example according to Def. 8. Then $d(x, \xi) = \|x - \xi\|_p \leq \delta$ and $\ell(f(\xi), y_\xi) - \ell(f(x), y) = \ell(f(\xi), f(x)) - \ell(f(x), f(x)) = \ell(f(\xi), f(x)) > \epsilon$. Therefore, $\xi$ is also an adversarial example according to Def. 2.

$\square$

## F   DETECTING ADVERSARIAL EXAMPLES

It is possible to detect adversarial examples using a simple threshold on the relative sharpness measure. We did not include a practical study of this since it would go beyond the scope of this paper. Developing a sound method requires more than fine-tuning the threshold. Moreover, this requires comparing the approach to a wide range of state-of-the-art detection methods, which would be a paper of its own. Therefore, we leave this interesting practical aspect for future work. Nonetheless, we provide preliminary results. We trained a decision stump on the sharpness of clean and adversarial samples on CIFAR-10 for WIDERESNET-28-4 using 5-fold cross-validation, which yields the following accuracies: $[0.92, 0.92, 0.93, 0.92, 0.92]$, i.e., adversarial examples can be detected with an average accuracy of 0.92 with little to no difference between the folds.

## G   REBUTTAL

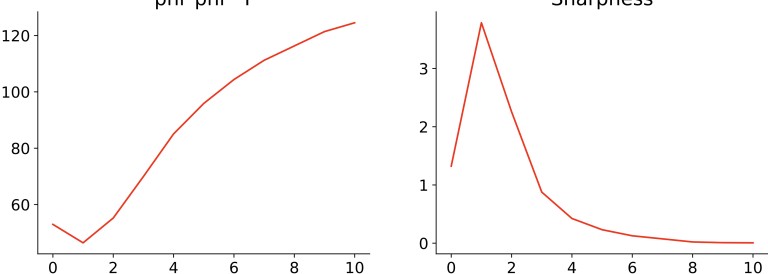

Figure 10: Here, we plot the squared norm of the hidden representation versus the sharpness.

