# OpenReview forum: "The Uncanny Valley: Exploring Adversarial Robustness from a Flatness Perspective"
_ICLR.cc/2025/Conference — ICLR 2025 Conference Withdrawn Submission_

### Official Review · Reviewer_3F1W · 2024-10-29

**Soundness:** 2
**Presentation:** 2
**Contribution:** 1
**Rating:** 3
**Confidence:** 4

**Summary:**

This paper reports an ``uncanny valley'' phenomenon and provides a theoretical connection between relative flatness and adversarial robustness.

**Strengths:**

This paper revealed an interesting phenomenon: the relative flatness first goes up and then down as the adversarial attack progresses.

**Weaknesses:**

1. The existing analysis and discussion are insufficient to demonstrate that the phenomenon observed in the paper is significant and well-motivated. Specifically, I have the following concerns:

     (1.a) The summary of conclusions in the existing literature on the relationship between landscape flatness and adversarial robustness is vague. Therefore, I have not established any intuition about this problem by reading the background part.

     (1.b) The author claimed multiple times that the valley is counterintuitive, which is very confusing to me. My questions are, intuitively, why adversarial examples should not lie in a flat region, and why the trend of flatness first increasing and then decreasing is unusual.

     (1.c) What are the takeaways from the experiments on adversarially trained models?


2.  The theoretical part is not well connected to the empirical observation and is not well explained.

     (2.a) The motivation of the theoretical part (Lines 406-412) is puzzling, especially the ``counterintuitive behavior''.

     (2.b) The theoretical results involve the Lipschitz condition concerning input, which is directly related to the adversarial robustness of the feature extractor.

     (2.c) The flatness analysis based on the penultimate layer is rather limited.

**Questions:**

1. If standard adversarial training is replaced with flatness-related adversarial training methods, such as Wu et al., 2020, how would the phenomena in Figures 4 and 5 change?


2.  Regarding the robust generalization problem, should we consider the flatness of the adversarial loss function, which refers to the maximal standard loss within a neighborhood?

---

> ### Author Response · Authors · 2024-11-21
>
> Dear Reviewer 3F1W,
>
> There seem to be multiple important misunderstandings that we wish to clarify. In addition, we give background knowledge on adversarial examples and flatness.
>
> ## 1. Significance of the Contribution
> We disagree that an investigation into the phenomenon of the uncanny valley and the relationship between flatness and adversarial robustness is not well-motivated. As our broad empirical evaluation shows, the uncanny valley phenomenon can be observed in a large variety of datasets and model architectures, including large language models. Our theoretical contribution shows that we can connect generalization and adversarial robustness through flatness of the loss, addressing a significant gap in the literature.
>
> ## 1.a Related Work
> The phenomenon of the uncanny valley is novel. To the best of our knowledge there exists no study of the flatness of the loss around adversarial examples in the literature. Existing work on adversarial attacks and defenses can be related to flatness and we discussed those in Sec. 2. It is often claimed that models that generalize better – as measured in terms of flatness – are better protected against adversarial attacks, which is at odds with the common  observation that adversarial robustness often leads to worse performance.Overall, this reflects a contradictory existing evidence of links between flatness and adversarial robustness.
>
> ## 1.b The Uncanny Valley Phenomenon is Counterintuitive
> Flatness has traditionally been associated with desirable properties like good generalization and robustness, which Schmidhuber and Hochreiter proposed in 1995. Given that adversarial examples are by definition, a consequence of poor generalization, one would expect that the sharpness of the model evaluated at the adversarial distribution is high. In layman's terms, the model lies in a sharp area. Hence, it is unintuitive that adversarial samples do not lead to a high sharpness.
>
> To reiterate: Since adversarial examples cross the decision boundary, it is to be expected that sharpness increases until the label flips. Although it has not been studied before, a decrease in sharpness afterwards is not unexpected. What **is unexpected**, however, is that adversarial examples then move into a **very flat** extended region, or valley. Since, intuitively, adversarial examples, which exploit vulnerabilities, should lie in sharp regions where the model is unstable. Thus, the fact that adversarial examples instead reside in very flat regions, as revealed by our empirical results, contradicts this intuition and raises critical questions about the sufficiency of flatness as a robustness metric. Our theory then complements these findings by showing that robustness is **indeed** related to flatness, but only around a given dataset.
>
> ## 1.c Conclusions from Experiments with Adversarial Training
> These experiments show that our theory is correct, because adversarial training increases flatness around clean examples. It also supports our crucial conclusion that while we can guarantee adversarial robustness around the training examples, flatness cannot inform on adversarial robustness away from it.
>
> To cite our manuscript: “If we attack these more robust models with stronger attacks, we again observe that the adversarial examples lie in flat regions (Fig. 4c and 5c). Hence, the uncanny valleys still exist and can be reached via stronger attacks.”

---

> > ### Author Response · Authors · 2024-11-21
> >
> > ## 2. Connection between Theory and the Uncanny Valley
> > While we can guarantee adversarial robustness around the training examples, flatness cannot inform on adversarial robustness away from it. So we are able to reconcile generalization and adversarial robustness around the training set through flatness, but we also show the limitations of flatness in explaining adversarial robustness: in the uncanny valley, there are adversarial examples that are very flat.
> > ## 2.a Counterintuitive Behavior
> > As discussed in answer to point 1.b, the theoretical results clarify the relation between relative sharpness, generalization, and adversarial robustness and partially reconcile generalization and adversarial robustness, while the uncanny valley phenomenon shows the limitations of flatness in explaining adversarial robustness.
> > ## 2.b Lipschitzness and Adversarial Robustness
> > By now, it has been well understood that Lipschitz continuous feature extractors are not enough to achieve adversarial robustness; for formal treatment, we refer to [1].
> > The main points are:  In order for Lipschnitzness to directly relate to adversarial robustness, the Lipschitz constant has to be close to $1$. While such neural networks exist, most neural networks have substantially larger Lipschitz constants. Our theoretical result indicates that some bound on Lipschitzness of the feature extractor is necessary to link flatness and adversarial robustness. It also shows, though, that the impact of Lipschitzness of the feature extractor is counter-balanced by relative sharpness.
> > ## 2.c Flatness in the Penultimate Layer
> > We kindly disagree that analyzing flatness in the penultimate layer is “rather limited”. Petzka et al., 2021 [2] have established that relative sharpness computed at the penultimate layer is sufficient to explain the network’s generalization. Moreover, our empirical results confirm that the uncanny valley phenomenon can be observable in all layers, as shown in our supplementary experiments (Appendix, Section B). Measuring sharpness at the penultimate layer is computationally efficient and captures the critical trends we report.
> >
> > ## Question 1
> >
> > Fig. 4 and 5 show that adversarial training increases the flat area around clean examples. Most likely this is due to adversarial training pushing the decision boundary further away from the clean examples and thus indirectly induce flatness. Explicitly inducing flatness, as proposed in Wu, et al. 2020 [3] should further extend these flat regions. Using our Prop. 5 it should be possible to provide an alternative guarantee on adversarial robustness for their adversarial weight perturbations that does not use the PAC-Bayes framework. Sharpness aware minimization has a similar effect and leads to adversarial robustness [4], further supporting our theoretical results.
> > ## Question 2
> > The flatness of the adversarial loss of a clean example can be translated to the robustness against perturbation around **the adversarial example where this loss is realized**, and thus cannot be directly related to the adversarial robustness of the clean example. An adversarial loss that would truly measure the maximum loss attainable in a $\delta$-radius around a clean example would constitute in itself a robustness guarantee. Such a loss should also imply flatness and thus it should be possible to link it to generalization through the same argument as in Sec. 6 of our manuscript. The adversarial loss function as used in practice, however, typically refers to the loss of an adversarial example found in a $\delta$-radius around the clean example and therefore does not provide an upper bound on the maximum loss attainable - it only provides a lower bound.
> >
> >
> >
> > We hope we could give sufficient background to understand the main contributions of our paper, if not, feel free to ask.
> >
> >
> > **References:**
> >
> > [1] Zhang, Bohang, et al. "Rethinking lipschitz neural networks and certified robustness: A boolean function perspective." Advances in neural information processing systems 35 (2022): 19398-19413.
> >
> > [2] Petzka, Henning, et al. "Relative flatness and generalization." Advances in neural information processing systems 34 (2021): 18420-18432.
> >
> > [3] Wu, Dongxian, Shu-Tao Xia, and Yisen Wang. "Adversarial weight perturbation helps robust generalization." Advances in neural information processing systems 33 (2020): 2958-2969.
> >
> > [4] Wei, Zeming, Jingyu Zhu, and Yihao Zhang. "Sharpness-Aware Minimization Alone can Improve Adversarial Robustness." The Second Workshop on New Frontiers in Adversarial Machine Learning.

---

> > > ### Comment · Reviewer_3F1W · 2024-11-24
> > >
> > > Thank you for providing additional clarifications. I acknowledge your theoretical contribution in establishing the connection between relative sharpness and adversarial robustness. Furthermore, the observations regarding adversarial training presented in Figure 4 are more compelling.
> > >
> > >
> > > However, I will not raise my rating due to two major concerns:
> > >
> > > 1. >What is unexpected, however, is that adversarial examples then move into a very flat extended region, or valley. Since, intuitively, adversarial examples, which exploit vulnerabilities, should lie in sharp regions where the model is unstable.
> > >
> > >     This claim appears to lack sufficient empirical or theoretical support and comes across as somewhat subjective. Given that adversarial training does not optimize for examples beyond the predefined budget, the resulting model is unlikely to account for adversarial examples that are farther away. I do not believe it is reasonable to assume the landscape around such “irrelevant” data points should necessarily be sharp.
> > >
> > > 2. >While we can guarantee adversarial robustness around the training examples, flatness cannot inform on adversarial robustness away from it.
> > >
> > >     As the authors acknowledge, the theoretical results do not explain the core aspect of the proposed “uncanny valley” phenomenon—namely, the flatness of the loss landscape far from the original data.
> > >
> > >
> > >
> > > In my view, the authors may benefit from reevaluating the emphasis placed on the “uncanny valley” phenomenon. Shifting the focus towards empirical results that are more directly tied to the theoretical framework would strengthen the overall contribution.

---

### Official Review · Reviewer_nqir · 2024-10-30

**Soundness:** 2
**Presentation:** 2
**Contribution:** 2
**Rating:** 3
**Confidence:** 4

**Summary:**

The paper investigates the vulnerability of deep learning models to adversarial attacks by examining the connection between adversarial examples and the flatness of the loss landscape with respect to the model's parameters. To assess the flatness, the authors use a metric termed "relative sharpness." Through empirical observations, the paper identifies a pattern that the relative sharpness values tends to be lower ---- suggesting the loss landscape is flatter ----  when it is measured on perturbed examples generated during later iterations of adversarial attacks. This flat landscape is named uncanny valley in the paper. The paper also presents a theoretical analysis that links the relative sharpness metric to the model's robustness against adversarial perturbations.

**Strengths:**

The paper reveals an interesting phenomenon that the model's loss landscape w.r.t its parameters becomes progressively flatter when measured on "stronger" adversarial examples—those generated during the later iterations of an adversarial attack.

This potentially brings valuable insights for understanding the vulnerability of DNNs to adversarial perturbations.

**Weaknesses:**

Although the theoretical analysis links the relative sharpness measure, $\kappa_{Tr}^{\phi}$ , with the robustness of the model, it does not directly explain the emergence of the uncanny valley phenomenon. The theoretical results in Proposition 5 and Corollary 6 indicate that a model with a flatter loss landscape would generally be more robust, which does not have an explicit connection with the uncanny valley phenomenon. Also the relative sharpness measure $\kappa_{Tr}^{\phi}$ in Proposition 5 and corrolary 6 is considered to be computed on unperturbed examples whereas the experiments apply it to adversarial examples.



The theoretical results lack experimental validation. In line 484, the paper asserts that a model with smaller relative sharpness will achieve both good generalization and adversarial robustness. However, this claim requires experimental verification to substantiate it.



I also have concerns regarding Definition 2 as proposed in this paper. In Definition 2, the notion of adversarial examples depends on the choice of the loss function, which may be arbitrary and potentially unrelated to the classifier $f$. This construction could be misleading, as an example $x$ that alters the output of $f$  would be considered adversarial under Definition 1 but be treated as non-adversarial under Definition 2 if $l$ is chosen as a constant function or as a function with a very small Lipschitz constant. Could you clarify why Definition 2 offers an improved framework for defining adversarial examples compared to Definition 1, as established in prior work?



Multiple statements and claims in the paper are unclear and lack theoretical support. For example:

- Line 053, " a vicinity of such an adversarial sample will be filled with similar adversarial examples". Could you clarify what is meant by "similar adversarial examples" in this context?
- Line 170, "adversarial examples are a consequence of non-smooth directions". What is "non-smooth direction"?
- Line 256, "as expected the loss and sharpness increase as the attack progresses". Why is this increase expected, and on what basis?
- Line 261, "meaning there are adversarial examples in a flat region for good performing". What does "good performing" refer to?
- Line 266, "Thus, the uncanny valleys truly extend away from the original example, supporting the intuition that adversarial examples live in entire subspaces of the input space." Could you provide a more detailed explanation of the causal reasoning behind this statement?

**Questions:**

See weaknesses.

---

> ### Author Response · Authors · 2024-11-21
>
> Dear Reviewer nqir,
>
> We appreciate your detailed review and the effort you put into evaluating our submission. However, we strongly disagree with your assessment of the paper’s soundness, presentation, and contribution. Below, we provide a point-by-point rebuttal to address your concerns and demonstrate that our work makes a significant and well-supported contribution to the field.
>
> ## Theoretical Contribution and the Uncanny Valley Phenomenon
> Indeed, in Prop. 5 and Cor. 6, relative sharpness is computed on the unperturbed training set. We then use the relation between flatness and robustness to random perturbation established in Petzka, et al [1] to prove _adversarial robustness_ around training examples. This offers a novel perspective by linking flatness of the loss curve, generalization, and adversarial robustness, addressing a crucial gap in existing literature.
> Our experiments confirm that for well-generalizing models, the area around training examples is flat. The fact that the flat area around training examples is increased through adversarial training (Fig. 4 and Fig. 5) **confirms our theory**. Moreover, [7] empirically observes what our theory predicts, that is, flatter regions are more robust to adversarial examples and generalize better.
>
> ## Definition of Adversarial Examples
>
> The definition of adversarial examples has always **implicitly** depended on the choice of loss function. In Szegedy [4], an adversarial example changes the prediction which is captured by Def. 2 using the assumption of locally constant labels and any common loss function with $\epsilon = 0$. In order to relate adversarial examples to **sharpness of the loss function**, it is **essential** to define adversarial examples wrt a loss function.
>
> The proposed Def. 2, which is a generalization of existing definitions, offers an improved framework over the literature for three reasons.
>   1. The existing definitions in the literature relate to the optimization problem of finding adversarial examples, but do not provide a general formal and practical definition. E.g., in Def. 1, only the closest point to the clean example that flips the label is considered an adversarial example, whereas a point that is just an $\epsilon$ further away would not be considered one. In practice, the attack algorithm does not converge to the optimum, so that the points found in the experiments are formally not adversarial examples. Def. 2 instead allows us to define the set of adversarial examples independent of an attack method.
>
>  2. As stated in Sec. 3, definitions of adversarial examples in the literature make several implicit assumptions (e.g., locally constant labels). These assumptions are reasonable in practice for image classification tasks, but should be made explicit. For example, in a regression task it is often the case that small changes to the input _do_ lead to large changes in the output. Thus, existing definitions would not apply here.
>
> 3. It is a general definition of adversarial examples, independent of the use case or data type. As we show in Appendix D, Def. 2 can be instantiated to cover existing definitions in the literature, including targeted attacks, such as Szegedy, et al [4], Papernot, et al [5], Carlini and Wagner [6], as well as untargeted attacks [6].

---

> > ### Author Response · Authors · 2024-11-21
> >
> > ## Clarification of Statements
> >
> > Your interpretation of specific statements (lines 053, 170, 256, 261, and 266) suggests a misunderstanding of our language or intent. To clarify:
> >
> >  - Line 053: "Similar adversarial examples" refers to inputs that yield the same incorrect prediction under similar perturbations, highlighting the clustered nature of adversarial subspaces.
> >  - Line 170: "Non-smooth directions" describe abrupt gradient changes, a known characteristic linked to adversarial vulnerability [2].
> >  - Line 256: The increase in loss and sharpness during early attack iterations is to be expected since the example crosses the decision boundary. Although it has not been investigated before, it is not unexpected that sharpness decreases once the decision boundary is crossed. What is unexpected, though, is the fact that sharpness drops to a very low value and remains low in an extended valley.
> >  - Line 261: "Good performing" refers to models with high accuracy on clean data, emphasizing that flat regions do not inherently imply adversarial robustness. This highlights an important aspect of our contribution: our theory proves that flatness around training examples guarantees adversarial robustness _for these training examples_, yet the uncanny valley phenomenon shows that we cannot infer adversarial robustness from flatness away from the training examples.
> >  - Line 266: The fact that adversarial examples extend away from the example shows that they are not located in small, compact regions close to a clean example. This supports the intuition that adversarial examples form entire subspaces [2] (e.g., as in the dimpled manifold [3]).
> >
> > These claims are grounded in well-established literature and are further supported by our empirical findings.
> >
> > ## Conclusion
> > We firmly believe that this paper makes a significant, sound, and timely contribution to the field of adversarial robustness, including for large language models. The combination of theoretical innovation and empirical validation addresses critical gaps in existing research.
> >
> > **References:**
> >
> > [1] Petzka H, Kamp M, Adilova L, Sminchisescu C, Boley M. Relative flatness and generalization. Advances in neural information processing systems. 2021 Dec 6;34:18420-32.
> >
> > [2] Tramèr, Florian, et al. "The space of transferable adversarial examples." arXiv preprint arXiv:1704.03453 (2017).
> >
> > [3] Shamir, Adi, Odelia Melamed, and Oriel BenShmuel. "The dimpled manifold model of adversarial examples in machine learning." arXiv preprint arXiv:2106.10151 (2021).
> >
> > [4] Szegedy, Christian, et al. Intriguing Properties Of Neural Networks. International Conference on LearningRepresentations, 2014.
> >
> > [5] Papernot, Nicolas, et al. "The limitations of deep learning in adversarial settings." 2016 IEEE European symposium on security and privacy (EuroS&P). IEEE, 2016.
> >
> > [6] Carlini, Nicholas, and David Wagner. "Towards evaluating the robustness of neural networks." 2017 ieee symposium on security and privacy (sp). Ieee, 2017.
> >
> > [7] Wei, Zeming, Jingyu Zhu, and Yihao Zhang. "Sharpness-Aware Minimization Alone can Improve Adversarial Robustness." The Second Workshop on New Frontiers in Adversarial Machine Learning.

---

> > > ### Comment · Reviewer_nqir · 2024-11-27
> > >
> > > Thank you for providing additional clarifications. I acknowledge that the empirical observations have the potential to offer valuable insights into the understanding of adversarial robustness. However, I will not raise my score as the main concerns remain insufficiently addressed:
> > >
> > > -  "Indeed, in Prop. 5 and Cor. 6, relative sharpness is computed on the unperturbed training set. We then use the relation between flatness and robustness to random perturbation established in Petzka, et al [1] to prove *adversarial robustness* around training examples. This offers a novel perspective by linking flatness of the loss curve, generalization, and adversarial robustness, addressing a crucial gap in existing literature. Our experiments confirm that for well-generalizing models, the area around training examples is flat. The fact that the flat area around training examples is increased through adversarial training (Fig. 4 and Fig. 5) **confirms our theory**. Moreover, [7] empirically observes what our theory predicts, that is, flatter regions are more robust to adversarial examples and generalize better."
> > >
> > > The authors confirm that "in Prop. 5 and Cor. 6, relative sharpness is computed on the unperturbed training set". This aligns with my understanding and highlights a critical disconnect that the theoretical analysis does not have a direct link with your empirical observations where the relative flatness are oberved w.r.t the perturbed samples.  So I guess the authors would also agree that the theoretical analysis does not explain the observed "uncanny valley" phenomenon.
> > >
> > > The authors claim that "This offers a novel perspective by linking flatness of the loss curve, generalization, and adversarial robustness" and that "Our experiments confirm that for well-generalizing models, the area around training examples is flat".  However the notion of generalization is never formally defined in the paper. Does the "generalization" in the paper refer to the robust generalization or the standard (clean) generalization? Is the "generalization" in the paper disscussed w.r.t a model trained by adversarial training or standard training?  In general, the paper's writing needs significant improvement to enhance clarity and rigor.
> > >
> > >
> > >
> > > - "The existing definitions in the literature relate to the optimization problem of finding adversarial examples, but do not provide a general formal and practical definition. E.g., in Def. 1, only the closest point to the clean example that flips the label is considered an adversarial example, whereas a point that is just an further away would not be considered one. In practice, the attack algorithm does not converge to the optimum, so that the points found in the experiments are formally not adversarial examples. Def. 2 instead allows us to define the set of adversarial examples independent of an attack method."
> > >
> > >
> > >
> > > This reply, along with related responses, fails to address my primary concern regarding Definition 2 that " In Definition 2, the notion of adversarial examples depends on the choice of the loss function, which may be arbitrary and potentially unrelated to the classifier $f$. This construction could be misleading, as an example $x$ that alters the output of $f$  would be considered adversarial under Definition 1 but be treated as non-adversarial under Definition 2 if $l$ is chosen as a constant function or as a function with a very small Lipschitz constant. "
> > >
> > > While I acknowledge that existing definitions (e.g., Definition 1) may not align perfectly with the problem settings explored in this paper, I disagree with the construction of Definition 2 (the proposed definition) where no restrictions are imposed on the choice of the loss function $l$.
> > >
> > >
> > >
> > > Regarding "Clarification of Statements," I appreciate the attempt to provide clarification. However, I notice that the manuscript remains unchanged, and the authors continue to use vague and unclear terminology, such as: "Similar adversarial examples", "Non-smooth directions",  "Good performing" , etc.  Such expressions lack precision and fail to meet the expected standards of rigor in academic writing.
> > >
> > >
> > >
> > > The current manuscript shows no intentions to improve either the writing or the technical aspects of the paper. Based on these reasons, I will keep my score unchanged.

---

### Official Review · Reviewer_xr3Q · 2024-11-01

**Soundness:** 3
**Presentation:** 3
**Contribution:** 2
**Rating:** 8
**Confidence:** 3

**Summary:**

This paper empirically analyzed the behavior of the loss function with respect to the parameters and adversarial examples. The loss surface initially becomes sharper as the attack progresses and the prediction is flipped. If the attack continues, the adversarial example often moves into a flat region (called an uncanny valley in the paper). This phenomenon appeared in various model architectures, including LLMs, datasets, and adversarially trained models.

**Strengths:**

There are abundant empirical results provided across different datasets and measures. The presentation of the overall paper is also concise.

**Weaknesses:**

Did not see a formal mathematical definition of "valley", but only a definition for measuring sharpness?

**Questions:**

How do you think the "uncanny valley" phenomenon would affect the developing defense techniques?

---

> ### Author Response · Authors · 2024-11-21
>
> Dear Reviewer xr3Q,
>
> Thank you for your positive assessment of our submission, in particular its soundness and concise presentation. Below, we address your concern and question.
>
> ## Formal Definition of "Valley"
> We do not provide a formal definition of valley, but use the term uncanny valley metaphorically to describe a set of observations: By valley we mean a constantly very flat region along the path of the adversarial example. This valley is uncanny in the sense that it is still very close to the original example, but confuses the predictor; it is also very flat and the predictor is very certain, yet the predictor is wrong.
>
> ## Practical Implications for Defense Techniques
> Our findings directly inform the development of adversarial defenses. Specifically, they reveal that adversarial examples tend to settle in unusually flat regions of the loss landscape post-label flip, which contradicts the assumption that flatness universally correlates with robust generalization. This insight underscores the need for hybrid defense strategies that combine flatness with smoothness constraints, as detailed in Section 6. Moreover, our preliminary experiments suggest the potential for leveraging relative sharpness as an effective tool for adversarial detection (Appendix E).
>
> ## Conclusion
> We hope these clarifications address your questions and highlight the significance of our contributions.

---

### Official Review · Reviewer_tZta · 2024-11-08

**Soundness:** 2
**Presentation:** 3
**Contribution:** 2
**Rating:** 3
**Confidence:** 3

**Summary:**

This paper empirically analyzes adversarial examples through the property of sharpness. As an adversarial attack gets progressively stronger, the loss surface first gets progressively sharper, and then flatter after the label is flipped. A measure of flatness can be used to theoretically bound the adversarial robustness for Lipschitz models, providing a theoretical connection between flatness and robustness.

**Strengths:**

The authors identify an empirical phenomena relating two areas, adversarial robustness and flatness of the loss landscape, potentially providing a path to increased adversarial robustness. They demonstrate this phenomena on a wide variety of models and datasets.

**Weaknesses:**

It is unclear what the bigger picture is with regards to the uncanny valley phenomena identified in this paper. In particular, consider the form of the trace of the Hessian on line 228. For adversarial examples if the feature extractor does not change much (since by definition we induce small perturbations when finding adversarial examples), then $\sum_{i=1}^d \phi_i^2$ is roughly constant, so the sharpness measure is essentially just $\sum_{j=1}^k \hat{y}_j(1-\hat{y}_j)$, from which all of the empirical results are easily explained. The relative sharpness is large when there are label probabilities that are not certain (far from 0 or 1). In the process of flipping the prediction from one label to another, the probabilities smoothly interpolate to a regime where the label probabilities are less certain, which explains why the relative sharpness first increases, and then decreases once the label is flipped. With this explanation the phenomena is not surprising, as this would just be a natural consequence of adversarial examples existing.

**Questions:**

- What practical effect does this empirical phenomena have? What can be done with this information?
- In the empirical results, do the adversarial examples have $\sum_{i=1}^d \phi_i^2$ roughly constant, so that the relative sharpness is mostly determined by $\sum_{j=1}^k \hat{y}_j(1-\hat{y}_j)$?

---

> ### Author Response · Authors · 2024-11-21
>
> Dear Reviewer tZta,
>
> Thank you for your detailed review and feedback. However, we strongly disagree with your assessment, particularly your conclusion that our findings are unsurprising and that the contributions of this work are insufficient. We firmly believe that our paper makes a significant and novel contribution to the field by establishing a robust empirical and theoretical connection between adversarial robustness and the flatness of the loss landscape, leading towards the discovery of the uncanny valley phenomenon. This phenomenon, which we rigorously demonstrate across multiple models and datasets, is far from trivial and provides novel insights into the geometry of adversarial examples. Moreover, our theoretical framework offers a novel perspective by linking generalization, relative sharpness, and adversarial robustness, addressing a crucial gap in existing literature. We contend that this paper is both sound and impactful, and thus merits acceptance.
>
> ## The Uncanny Valley Phenomenon
>
> You correctly point out that the sharpness for an example depends on the confidence of the predictor. That is, the form of the trace of the Hessian in the penultimate layer we derived reveals that very high-confidence predictions are usually flat. This does not explain the observations, though. Indeed, confidence is directly related to the distance to the decision boundary. Consider a binary classification task using logistic regression and CE loss. The prediction goes to $1$ for $w^⊤x+b\rightarrow\infty$ and $0$ for $w\top ⊤ x+b\rightarrow −\infty$. Keeping $w,b$ fixed, this is achieved only when the distance of $x$ to the decision boundary $d=\frac{|w^⊤x+b|}{|w|}$, grows to infinity. High confidence predictions are in regions far away from the decision boundary, where predictions are stable, i.e., flat regions. The flatness measure exactly captures that. A well-trained network is flat around training points, but not perfectly flat. The curious phenomenon is that, when moving away from the training point during the attack, sharpness first increases around the decision boundary (as it should be) and then drops to an extremely low value. This low value persists as the attack progresses. Interestingly, the adversarial example never comes close to any other decision boundary again. Thus adversarial examples are not tiny regions close to clean examples in the otherwise reasonable decision boundary, but rather lie in a broad uncanny valley. We demonstrate this across diverse datasets and architectures (including LLMs), namely that adversarial examples often settle in these unusually flat regions of the loss landscape post-label flip (see Section 5, Figure 2). This behavior persists even under adversarial training, which traditionally aims to eliminate such vulnerabilities.
>
> ## Practical Implications
>
> The goal of our paper is to reconcile flatness, which is related to generalization, with adversarial robustness, since it has been put into question whether generalization and adversarial robustness are at odds [1]. Methods to improve adversarial robustness, like randomized smoothing and SAM indicate that flatness improves adversarial robustness. Together with the theoretical link of flatness to generalization, one could come to the conclusion that flatness implies that a predictor is correct. Our empirical results, however, show that adversarial examples live in wide, very flat regions. This contradicts the previous intuition that flat regions imply good performance. Our theoretical contribution makes this point more precise: flatness around training examples indeed implies both good generalization and adversarial robustness **on those examples**, but does not guarantee adversarial robustness on unseen data points.
> While the focus of our paper is to investigate this relationship, we did perform a preliminary experiment on whether flatness can be used to detect adversarial examples. We show that indeed a detection accuracy of $>90%$ is possible using a simple threshold.

---

> > ### Author Response · Authors · 2024-11-21
> >
> > ## Hessian Trace and Sharpness Measure
> >
> > According to our observations, the representations $\phi$ do change considerably even for small changes in the input. This is not unexpected, since the Lipschitz constants of the feature extractor are typically large. This can also be observed indirectly from Fig. 3: we observe that the distance in feature space is growing substantially more rapidly than in input space (between one and three orders of magnitude). One can also directly measure the norm of the representation and observe that it grows substantially during the adversarial attack (see Fig. 10 in the updated manuscript).
> >
> > ## Conclusion
> > In summary, your concerns, while valuable, seem to stem from a misinterpretation of our core contributions. We provide a theoretical (not only empirical) connection between flatness and adversarial robustness. Our goal is to pinpoint the applicability of flatness to adversarial robustness, not to provide a method for increased robustness.The uncanny valley is not a trivial phenomenon but a significant insight into the geometry of adversarial examples. We hope this response clarifies your concerns and convinces you of the merit and originality of our work.
> >
> >
> >
> > **References**
> >
> > [1] Tsipras D, Santurkar S, Engstrom L, Turner A, Madry A. Robustness may be at odds with accuracy. arXiv preprint arXiv:1805.12152. 2018 May 30.

---

> > > ### Comment · Reviewer_tZta · 2024-11-27
> > >
> > > Thanks for your response.
> > >
> > > I agree that the adversarial examples not approaching another decision boundary is interesting. I also found the added figure (Fig 10) to be interesting, especially as the norm of the representation is anti-correlated with the sharpness measure. This seems to suggest that the relative sharpness is dominated by the prediction probabilities. As a result, it remains unclear to me what aspects that are attributed to relative sharpness here could not instead be attributed to the prediction probabilities alone.

---

### Note · Authors · 2024-12-03

I have read and agree with the venue's withdrawal policy on behalf of myself and my co-authors.